# Nascent polypeptide-Associated Complex and Signal Recognition Particle have cardiac-specific roles in heart development and remodeling

**Analyne M. Schroeder**○*, **Tanja Nielsen, Michaela Lynott**○, **Georg Vogler**○, **Alexandre R. Colas**○, **Rolf Bodmer***

Development, Aging and Regeneration Program, Center for Genetic Disorders and Aging Research, Sanford Burnham Prebys Medical Discovery Institute, La Jolla, California, United States of America

* aschroeder@sbpdiscovery.org (AMS); rolf@sbpdiscovery.org (RB)

**Data Availability Statement:** All relevant data are within the manuscript and its Supporting Information files.

## Abstract

Establishing a catalog of Congenital Heart Disease (CHD) genes and identifying functional networks would improve our understanding of its oligogenic underpinnings. Our studies identified protein biogenesis cofactors Nascent polypeptide-Associated Complex (NAC) and Signal-Recognition-Particle (SRP) as disease candidates and novel regulators of cardiac differentiation and morphogenesis. Knockdown (KD) of the alpha- (*Nacα*) or beta-subunit (*bicaudal*, *bic*) of NAC in the developing *Drosophila* heart disrupted cardiac developmental remodeling resulting in a fly with no heart. Heart loss was rescued by combined KD of *Nacα* with the posterior patterning *Hox* gene *Abd-B*. Consistent with a central role for this interaction in cardiogenesis, KD of *Nacα* in cardiac progenitors derived from human iPSCs impaired cardiac differentiation while co-KD with human *HOXC12* and *HOXD12* rescued this phenotype. Our data suggest that *Nacα* KD preprograms cardioblasts in the embryo for abortive remodeling later during metamorphosis, as *Nacα* KD during translation-intensive larval growth or pupal remodeling only causes moderate heart defects. KD of SRP subunits in the developing fly heart produced phenotypes that targeted specific segments and cell types, again suggesting cardiac-specific and spatially regulated activities. Together, we demonstrated directed function for NAC and SRP in heart development, and that regulation of NAC function depends on *Hox* genes.

## Author summary

Identifying novel genes involved in cardiac development could help patients with Congenital Heart Disease through improved understanding of the developmental missteps, more precise patient diagnosis, and invention of targeted medical interventions. We identified protein biogenesis cofactors Nascent polypeptide Associated Complex (NAC) and Signal Recognition Peptide (SRP) to be involved in cardiac specific roles during development. Disruption of NAC and SRP subunits led to distinct and cell targeted disruptions in

**Funding:** This work was funded by grants from the National Institutes of Health R01 HL054832 to RB and F32 HL131425-01 to AMS. Support also includes funding from The Department of Defense (W81XWH-21-1-0159) to AMS. The funders had no role in study design, data collection and analysis, decision to publish, or preparation of the manuscript.

**Competing interests:** The authors have declared that no competing interests exist.

the heart, which would be unexpected if NAC and SRP merely had generic cellular function. Specifically, in flies, knockdown (KD) of the alpha- subunit of NAC, *Nacα*, led to an adult fly with no heart that could be rescued by co-KD with the posterior patterning *Hox* gene *Abd-B*, indicating a critical developmentally relevant genetic interaction, and not merely a generic protein biogenesis defect. This interaction was recapitulated in human Cardiac Progenitors, whereby human *NACA* KD redirected progenitor differentiation away from cardiomyocyte and toward fibroblast fates, which was rescued by concomitant *Hox* gene KD. Lastly, *Nacα* activity was required at specific times during development, and depending on when *Nacα* was knocked down, the resulting adult heart could be mildly or severely malformed. Thus, the work presents a new class of genes involved in protein biogenesis that display tissue- and temporal-specific activities that are crucial for proper heart development.

## Introduction

Congenital Heart Disease (CHD) is characterized by structural malformations of the heart present at birth caused by deviations from the normal course of cardiogenesis [1]. Genetics is a critical driver of CHD [2, 3]. Chromosomal anomalies as well as variants in genes involved in heart development have been identified in CHD patients [4]. These genetic features and disease presentations are heritable and cluster in families [5, 6]. Identifying the genes associated with disease helps piece together genetic networks that could uncover mechanisms underlying pathogenesis. Approximately 400 genes have been implicated in CHD [2], some that cluster within defined pathways, which permits a genetic diagnosis for approximately 20% of CHD patients. However, this leaves the vast majority of CHD cases with unknown genetic origins [3]. Therefore, expansion of the genetic data pertinent to CHD, such as functional analysis of genes with variants of uncertain significance (VUS), would advance our understanding of the disease and may offer, in the future, a diagnosis and targeted treatment for CHD patients. A better understanding of additional genetic risk factors and patient-specific combinations of such factors is aided by identification of candidate disease genes through patient-specific genomics, such as whole genome sequencing (WGS), followed by their evaluation in cardiac developmental platforms and assays from various genetic model systems [7, 8]. These validation efforts can accelerate candidate gene identification and focus on new potentially pathogenic genes, including genes located within larger genomic anomalies such as *de novo* Copy Number Variants [9].

Previous data suggest that the human gene Nascent polypeptide Associated Complex-alpha (*NACA*) is a candidate CHD gene that could provide novel insights into biological pathways in cardiac morphogenesis and pathogenesis. Using a GWAS approach, *NACA* was located within a genomic locus associated with increased myocardial mass [10], while Whole Exome Sequencing in families with Tetralogy of Fallot identified a single nucleotide polymorphism within *NACA* [11]. *NACA* is a highly conserved *alpha* subunit of a heterodimeric complex called Nascent polypeptide Associated Complex (NAC). Along with its heterodimeric partner, NAC-beta *(NACβ/BTF3)*, NAC is one of several chaperones found near the ribosome exit tunnel that bind to select emerging nascent polypeptides [12]. NAC-ribosome complexes facilitate transport of nascent polypeptides to the mitochondria as has been demonstrated in yeast [13–15]. At the ribosome exit site, NAC gates the activity of other nascent polypeptide chaperones. For example, NAC enhances the fidelity of Signal Recognition Particle (SRP) binding to only those nascent polypeptides destined for import to the Endoplasmic Reticulum (ER) [16–18].

Depletion of NAC leads to promiscuous binding of SRP onto nascent polypeptides, drawing mistargeted ribosome-nascent polypeptide complexes to the ER for aberrant insertion into the membrane or secretion. *NACA's* function as part of NAC therefore regulates the localization and posttranslational quality control of proteins in the cell.

In *Drosophila*, NAC plays an important role in translational regulation critical for embryonic development [19, 20]. Fly homologs of NAC subunits, *Naca* and bicaudal *(bic)*, were shown to repress protein translation of the posterior patterning gene oskar (*osk*) in anterior regions of the embryo, which was despite association of *osk* mRNA with polysomes, usually indicative of active translation. Restricting OSK protein translation and accumulation to the posterior pole of the embryo is critically required for patterning the posterior body plan [21]. Depletion of either *Naca* or *bic*, expands OSK protein localization anteriorly resulting in a bicaudal phenotype, where the embryo develops with mirror-image duplication of the posterior axis [19, 22]. These studies suggest that within the developing embryo, NAC appears to regulate the expression of select proteins for proper spatial distribution. NAC could have a similar role in the timing and spatial targeting of translation within specific tissues.

*Naca* has been demonstrated to be critical for development of several tissues using various model organisms. In mouse, the *Naca* subunit can function as a transcriptional coactivator regulating bone development [23–25] and hematopoiesis in zebrafish [26]. In vertebrates, a skeletal muscle- and heart-specific variant of *Naca* has been associated with myofibril organization in zebrafish [27] and muscle and bone differentiation in mouse [28–32]. Recent studies in the fly showed that *Naca* knockdown (KD) specifically in the heart led to a 'no adult heart' phenotype [10], suggesting that *Naca* could play a role in sarcomeric biogenesis, but its exact role in cardiac development had been unclear.

Here, we provide evidence for a cardiac developmental role for *NAC and SRP* in *Drosophila* and *NACA* in human Multipotent Cardiac Progenitors (MCPs). In flies, cardiac KD of *NACα* (mentioned above) and *bic* throughout development led to complete loss of the heart. We demonstrate that this phenotype is dependent on the timing of *NACα* KD, which requires KD during both embryonic heart development and pupal cardiac remodeling for a complete loss of the adult heart. KD in embryos only led to cardiac constriction and loss of the terminal chamber, while retaining heart structures in the anterior segments. KD of *Naca* only during pupal stages did not affect adult heart structure. This suggests that NACα KD primes cardiac cells already in the embryo for aberrant responses to morphogenic cues during later developmental stages. Persistent *Naca* KD during pupal stages remained required for complete loss of the adult heart. Consistent with this idea, *NACα* KD throughout cardiac development induced ectopic expression of the posterior patterning *Hox* gene Abdominal-B (*Abd-B*) into anterior regions of the remodeling heart during pupation. Concurrent KD of *NACα* and *Abd-B* significantly rescued the heart. This interaction between *Naca* and *Hox* gene was recapitulated in MCPs, whereby *NACA* KD led to deviations in progenitor cell differentiation away from cardiomyocytes and toward fibroblast cell fates, which was reversed with combined KD of *NACA* and *Hox* genes *HOXC12* or *HOXD12*. Because NAC associates and influences the activity of SRP, we tested the effects of individual SRP subunit KD on the fly heart. Interestingly, KD of individual SRP subunits produced cardiac phenotypes, some of which were distinctly different from *NACα* KD. These results suggest specific roles for ubiquitously expressed protein biogenesis factors NAC and SRP in heart morphogenesis, in part through alterations in the *Hox* gene expression. Translational regulation adds to our growing knowledge of biological pathways that may specifically contribute to cardiac pathogenesis leading to CHD.

## Results

### Knockdown of *Nacα* and *bicaudal* in the heart throughout development results in complete absence of the adult heart

The Nascent Polypeptide Complex (NAC) is a heterodimeric complex made of an alpha (NACA/*Nacα)* and beta (NACβ/BTF3/bicaudal) subunit that binds ribosomes to influence translation, protein folding, and transport of select nascent polypeptide chains. We investigated the role of the NAC complex, focusing on the alpha subunit, in the development and function of the *Drosophila* heart. Consistent with previously published data [10], knockdown (KD) of *Nacα* by RNAi (KK109114) using the Hand4.2-GAL4 driver [33] that is active in the heart tube, surrounding pericardial cells and wing-hearts throughout development, led to a 'no-heart' phenotype in adults when stained with phalloidin and the heart specific collagen, *pericardin* (**Fig 1A and 1B**). Similarly, KD of the β-subunit, *bicaudal* (KK104718), using Hand4.2-GAL4 led to a 'no-heart' phenotype (**Fig 1C**). Despite *Nacα*, or *bic* KD throughout heart development leading to the absence of hearts in adults, the posterior aorta and heart tube were still present in controls, *Nacα* KD, and *bic* KD during early pupae (<20hr After Puparium Formation, APF) as captured by *in vivo* imaging of fluorescently labeled hearts (**Figs 1D–1F and S1A**). We filmed and followed cardiac remodeling through pupation [1–3]. Normally, the heart undergoes remodeling through trans-differentiation of the larval aorta into the adult heart tube (**Figs 1D** and 3A **and S1 Video**). However, after 20 hours of puparium formation (APF) the hearts of cardiac *Nacα* KD flies began to disappear as cardiac remodeling progressed (**Fig 1E and S2 Video**). The space at the dorsal midline usually taken up by the heart, is now filled in with fat cells, indicating that the heart undergoes complete histolysis during remodeling. A similar course of events is observed when *bicaudal* is knocked down in the heart (**Fig 1F and S3 Video**). We also stained *Nacα* KD hearts with phalloidin just prior to cardiac remodeling (about 26–28 AFP), where heart tubes were present, but were narrower with disorganized actin filament arrangement (**S1B Fig**). KD with a second *Nacα* RNAi line (GD36017) led to formation of an adult heart, however, function was severely impaired and cytoskeletal structures were highly disorganized (**S2A and S2B Fig**). A second *bicaudal* RNAi line produced a no heart phenotype in adults (**S2C Fig**).

Using other cardiomyocyte-specific drivers, conferring different developmental expression patterns than Hand4.2-GAL4 to reduce *Nacα* levels, *i.e. tinHE*-GAL4 and *tinCΔ4*-GAL4, did not produce severe heart loss but altered several parameters of heart function and structure, as measured by SOHA (see methods; **Fig 2A and 2B**). Diastolic diameters were decreased using both drivers (**Fig 2C**) without a change in systolic diameter (**Fig 2D**), and consequently reducing contractility, as measured by diminished fractional shortening, FS (**Fig 2E**). No changes were detected in heart period (**Fig 2F**) and diastolic intervals (**Fig 2G**). KD of *Nacα* with *tinCΔ4*-GAL4 caused a slight reduction in systolic interval, which is the duration of active heart contraction and relaxation (**Fig 2H**). Phalloidin staining of the adult hearts with reduced *Nacα* expression showed gaps between the circumferential myofibrils and sarcomeric disorganization compared to controls, consistent with reduced contractility (**Fig 2I**). These results suggest that at certain expression thresholds and timing where heart development is not completely diverted, *Nacα* may play a role in functional and structural aspects of the heart as well. Because *Hand*4.2-GAL4 drives expression in both cardiac cells and the pericardial nephrocytes, we tested the effect of *Nacα* KD in pericardial cells only using *Dot*-GAL4 [34] to assess their contribution to the overall heart phenotype and reflect on the tissue specificity of *Nacα* KD effects. (**Fig 2B–2H**). KD in the pericardial cells did not produce significant changes in heart function, except for a slight increase in systolic diameter that did not change contractility (**Fig 2D and 2E**). The organization of the circumferential myofibrils as stained by

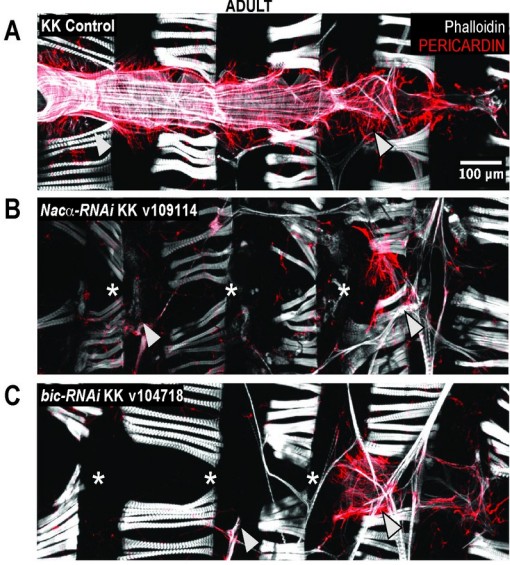

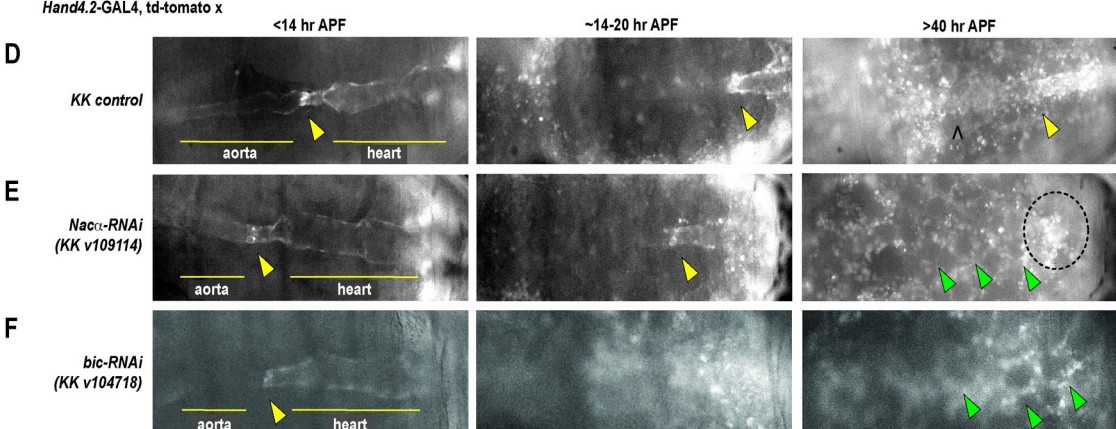

**Fig 1. Knockdown (KD) of either subunit of Nascent polypeptide Associated Complex led to complete histolysis of the heart in adult flies.** *Hand4.2*-GAL4 driving heart specific expression of **A, KK** control **B,** *Nacα*-RNAi (KK v109114) or **C,** *bicaudal*-RNAi (KK v104718). KD of *Nacα* or *bicaudal* led to an absent heart in adult flies. * indicates absence of the heart tube. Arrowheads point to remnants of alary muscles that normally attach to the heart for structural support (anterior-left). Pericardin, a heart-specific collagen is largely absent, except for remnants in the posterior end. **D,** Representative still images from *in vivo* imaging movies (S1–S3 Videos) of remodeling pupal hearts from control, *Nacα*, and *bicaudal* KD flies. During the first hours of pupation, the internal valves (yellow arrowheads) are visible which separate the larval aorta (anterior) and the heart (posterior) which we use as a landmark through remodeling. At about 14-20hr APF, these internal valves are present but the aorta is more difficult to visualize as the heart transdifferentiates. After 40hr APF, the remodeling adult heart tube is visible in controls with identifiable ostia structures (marked by a ^). In *Nacα* and *bicaudal* KD hearts, no heart tube is visible and the area is filled in by rounded fat cells (green arrowheads). The area of the embryo with fluorescent signal (circled) are remnants of histolyzed cardiomyocytes that slowly disperse and weaken in intensity. APFs are approximate due to developmental delays caused by reduced ambient temperatures in the microscope room.

phalloidin was unaltered compared to control (**Fig 2I**), suggesting that KD of *Nacα* in extra-cardiac cells, such as pericardial cells, contribute minimally to the overall cardiac phenotype.

## *Nacα* genetically interacts with the *Hox* gene *Abd-B* in heart development

We sought to better understand the mechanisms driving complete histolysis of the heart tube during metamorphosis. It is well established that during normal cardiac remodeling the posterior most segment of the larval heart (abdominal segments 6–7) that expresses the *Hox*

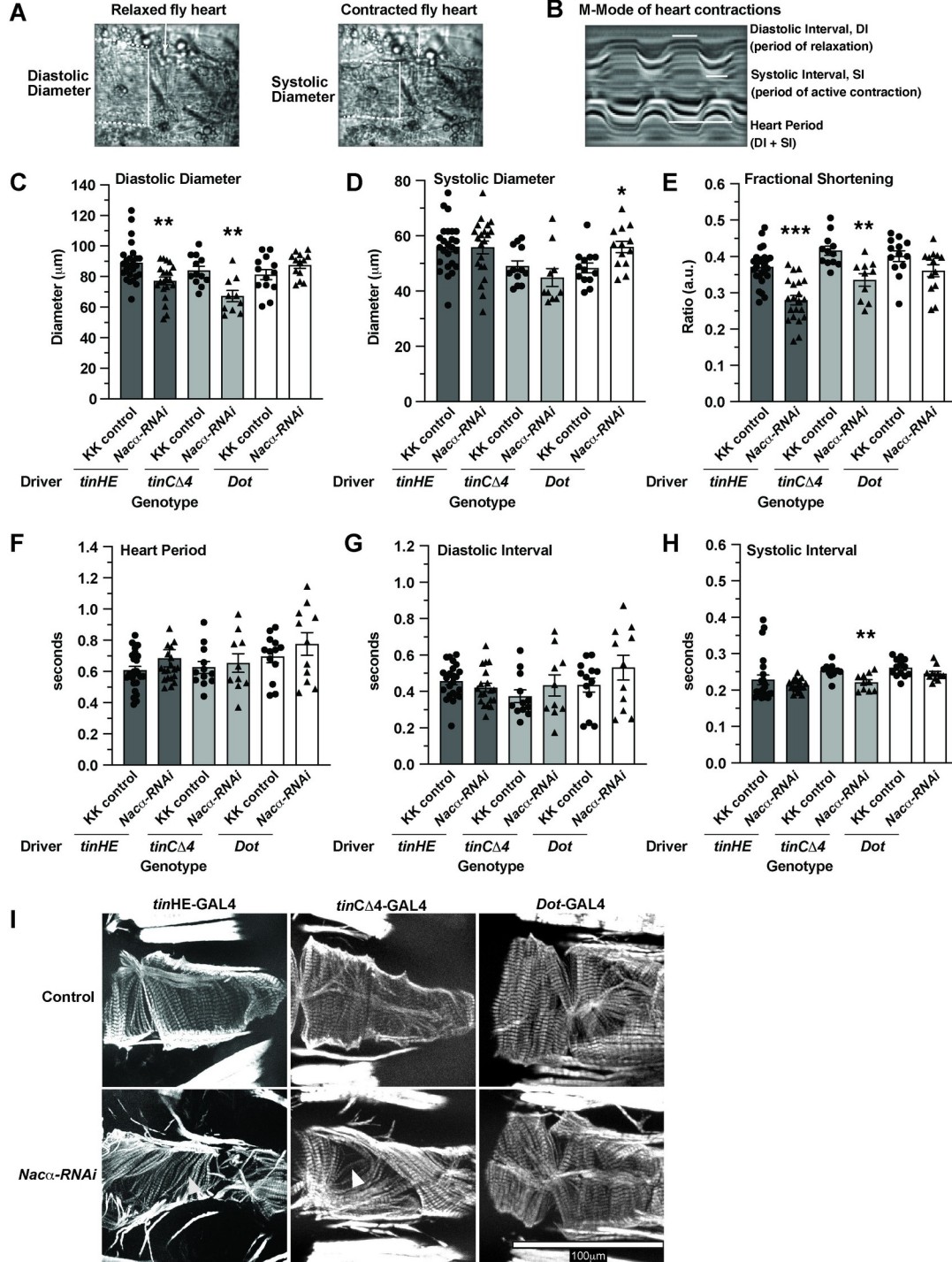

**Fig 2. Effect of *Nacα* knockdown (KD) using various GAL4 drivers on heart function and structure. A,B** Structural and functional parameters measured by SOHA to assess the fly heart. Dotted lines indicate the heart tube borders. White solid vertical lines indicate the diameters of the heart, while horizontal lines indicate the duration of contraction/relaxation being measured. **A,** Diastolic Diameter measures the heart diameter when it is fully relaxed, while systolic diameter measures the heart diameter when it is fully contracted. **B**, Motion-mode (m-mode) of the heart for temporal resolution of heart movement. Diastolic Interval measures the duration during which the heart is non-contractile, which occurs in this denervated fly preparation. Systolic Interval measures the duration that the heart is in active contraction and relaxation. **C**, Both *tinHE*-GAL4 and *tinCΔ4*-GAL4 cardiac drivers reduced Diastolic Diameter when used to KD *Nacα* expression (KK v109114). *Dot*-GAL4 pericardial cell driver had no effect on diastolic diameter. **D,** Cardiac drivers had no significant effect on systolic diameters,

while *Dot*-GAL4 increased systolic diameter slightly, indicating mild systolic dysfunction. **E,** Fractional Shortening is significantly decreased using both *tinHE*-GAL4 and *tinCΔ4*-GAL4 driver, while *Dot*-GAL4 had no effect. No changes in Heart Period **F,** or Diastolic Interval **G,** were detected with either cardiac drivers or *Dot*-GAL4. **H,** A slight reduction in systolic interval was detected when *Nacα*-RNAi was driven with *tinCΔ4*-GAL4. **I,** Adult fly hearts were stained with phalloidin to examine cytoskeletal structures following *Nacα* KD using various tissue drivers. Compared to controls that display well- and tightly organized circumferential fibers that drive heart contractions, both *tinHE*-GAL4 and *tinCΔ4*-GAL4 drivers led to alterations in the organization of fibers. White arrowheads point to gaps in the fibers in KD samples consistent with the observed reductions in fractional shortening. KD of *Nacα* expression using *Dot*-GAL4 did not cause alterations in circumferential organization.

segmentation gene Abdominal-B (*Abd-B*) histolyzes and is no longer present in adult hearts (**Fig 3A**) [35–38]. Abdominal segment 5 in the larvae that expresses the *Hox* segmentation gene abdominal-A (*abd-A*) remodels to become the adult terminal chamber, while the larval aorta (abdominal segment 1–4) that expresses *Hox* gene *Ultrabithorax (Ubx)* remodels to become the adult heart proper. We postulated that KD of *Nacα* throughout the heart could result in the misexpression of *Abd-B* leading to complete histolysis of the heart. First, we examined whether overexpression of *Abd-B* throughout the heart would result in similar phenotypes as with *Nacα* KD. Previously published data overexpressing *Abd-B* using an early pan-mesodermal driver (*twist*-GAL4) led to severely diminished embryonic muscle and heart development [39]. Using the cardiac-specific driver Hand4.2-GAL4 to overexpress *Abd-B*, the heart was completely absent in adults (**Fig 3B, right**), however a heart tube was present in larvae and in early pupae (24 hours APF, **Fig 3D**), similar to control (**Fig 3C**) *Nacα*-RNAi (**Fig 3E**) and *bic-RNAi* phenotypes (**Fig 1**). These results suggest that *Abd-B* expression and developmental activity are highly temporally controlled, and drive cardiomyocyte histolysis only during cardiac remodeling at pupal stages.

We then examined whether KD of *Nacα* leads to ectopic expression of Abd-B protein in the early pupal heart just prior to remodeling (APF 26–28). In controls, Abd-B protein is undetected in the anterior regions of the early pupal heart tube (**Fig 3C**), but detected in the posterior segments of the embryo. Overexpression of *Abd-B* using *Hand*4.2-GAL4 led to increased ABD-B protein accumulation throughout the early pupal heart tube and pericardial cells, which was restricted to the nuclei (**Fig 3D**). When *Nacα* was knocked down in the heart, we detected ABD-B protein expression throughout the larval/early pupal heart tube, within the myocardial nuclei as well as in the cytoplasm (**Fig 3E**). In orthogonal sections, there is ectopic ABD-B staining centrally, possibly within the lumen of the heart, either secreted from cardiomyocytes or bound to circulating hemocytes. These data suggest that KD of *Nacα* in the heart led to ectopic expression of ABD-B within the cell, as well as, anteriorly throughout the heart tube. Because overexpression of *Abd-B* using *Hand*4.2-GAL4 led to heart histolysis during remodeling, we speculate that ectopic ABD-B expression induced by *Nacα* KD could be leading to the observed histolysis of the entire heart during remodeling.

Because *Nacα* KD led to ectopic expression of ABD-B, we tested whether the no-heart phenotype produced by *Nacα* KD could be rescued by concurrent KD of *Abd-B*. KD of *Abd-B* alone led to largely intact hearts except for posterior ends that were more dilated and prominent compared to controls suggesting incomplete histolysis during cardiac remodeling, as expected (**S3 Fig**). KD of both *Nacα* and *Abd-B* using the Hand4.2-GAL4 driver led to a remarkable rescue of the adult heart, restoring formation of most of the circumferential myofibrils in the anterior regions (**Fig 3F and 3G**). These hearts had reduced diastolic diameters, significantly reduced fractional shortening (**Fig 3H**). This rescue was not due to titration of GAL4 protein by competing UAS sites, since a no-heart phenotype was still produced when *Nacα*-RNAi was combined with another UAS that carries a UAS-Val10-GFP construct (**Fig 3I**). These rescue data suggest a genetic interaction between *Nacα* and *Abd-B*.

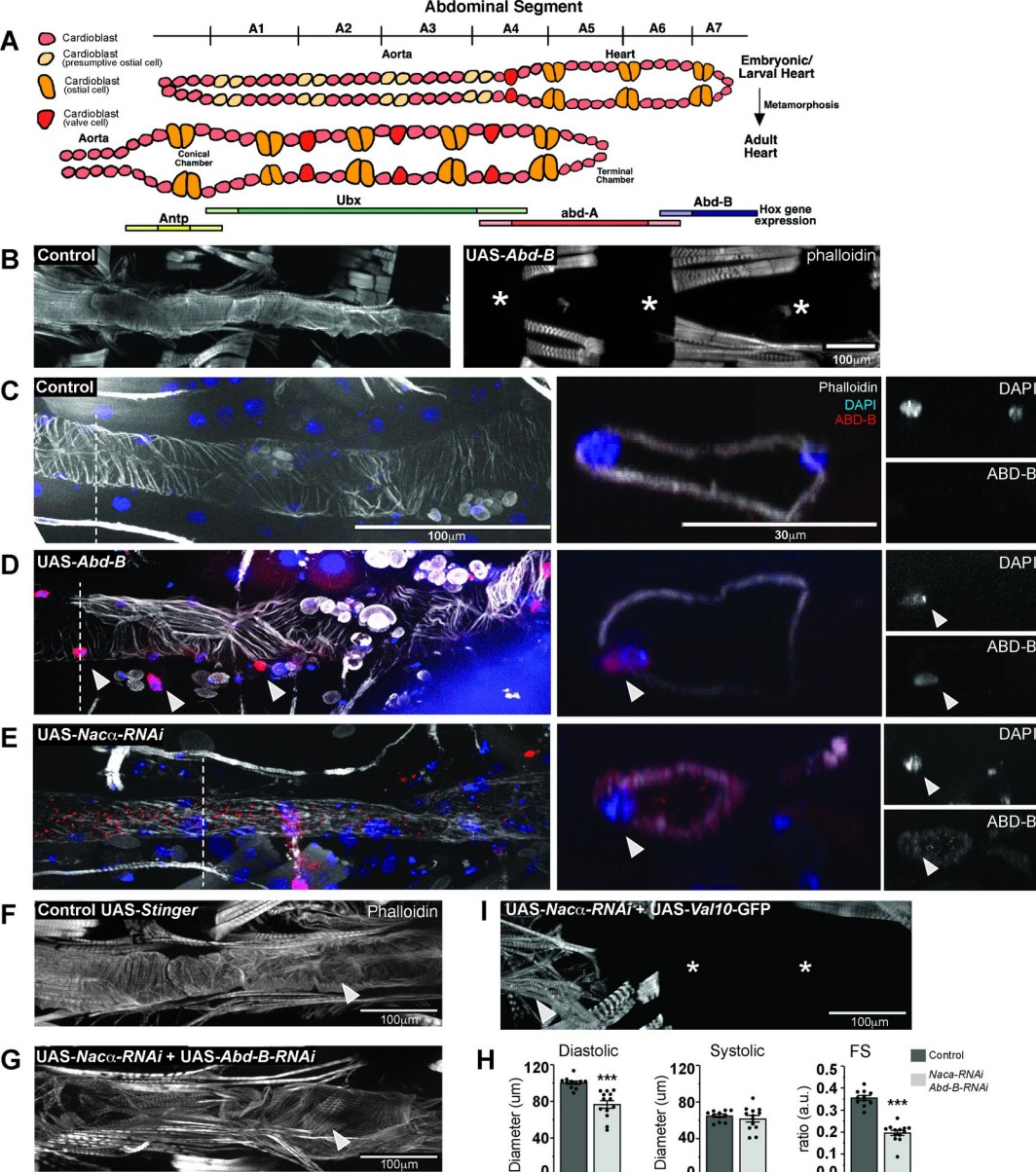

**Fig 3. *Nacα* and the *Hox* gene Abdominal B (*Abd-B*) genetically interact in the heart. A,** The embryonic/larval fly heart remodels into adult structures through trans-differentiation of the *Ubx* and *abd-A* expressing cardiomyocytes into the adult heart and terminal chamber, respectively. The cardiomyocytes of the posterior larval heart that express *Abd-B* (abdominal segment 6–7) histolyze and are absent in the adult heart. **B,** Overexpression of *Abd-B* in the heart using the cardiac driver *Hand4.2*-GAL4 led to complete absence of the adult heart, which in early stages of pupation was still present (see **D**). * indicates absence of the heart tube. **C-E.** Immunohistochemistry of pupal hearts 26–28 hours After Puparium Formation (APF). Dashed lines mark the region of the heart where orthogonal cross-sections were taken to examine *Abd-B* and DAPI expression in the nuclei. Images of the cross-section of cardiomyocytes and nuclei are displayed in the right panel. Arrowheads indicate either cardiac and pericardial nuclei that are both ABD-B and DAPI positive. **C,** In controls, phalloidin stained the circumferential fibers of the pupal aorta and heart. ABD-B staining was not detected in the cardiomyocyte nuclei within aorta and anterior heart segments but, ABD-B is stained in the posterior segments of the embryo. **D,** Overexpression of *Abd-B* using *Hand4.2*-GAL4 resulted in strong ABD-B staining in the nuclei (as indicated by the arrowheads) throughout the pupal heart and pericardial cells prior to remodeling. Cross section clearly shows that ABD-B is localized in the nucleus. **E,** Knockdown of *Nacα* in the heart resulted in ectopic ABD-B expression throughout the heart tube. Staining was present within the nuclei (marked by DAPI), cytoplasm and heart lumen (see orthogonal sections, right). **F-H.** Concurrent knockdown of *Nacα and Abd-B* in the heart, **G,** led to rescue of heart tube formation, with visible circumferential fibers (arrowheads) albeit less well-organized than controls. **H,** Rescued hearts were smaller in diastolic diameter with no change in systolic diameter resulting in reduced Fractional Shortening (FS). This effect was not due to titration of GAL4 onto 2 UAS sites, as combination of UAS-*Nacα-RNAi*

with UAS-Val10-GFP still produced a no heart phenotype. **I**, Arrow indicates remnants of the ventral longitudinal muscle in the anterior end of the abdomen.

Rescue by *Abd-B-RNAi* was not observed in the wing, where *Nacα* KD leads to fluid filled blisters in almost all flies, likely due to defects in wing-heart function (**S4 Fig**). Co-expression of *Nacα*-RNAi and *Abd-B*-RNAi did not improve the wing blisters nor reduce their penetrance (**S4C and S4D Fig**), while *Abd-B* KD by itself did not affect wing heart function (**S4A and S4B Fig**) or cause heart loss (**S3 Fig**), suggesting that the interaction and rescue may be cardiac tissue specific. Furthermore, KD of the *Hox* gene *abd-A* in combination with *Nacα-RNAi* did not rescue the loss of the heart (**S5A Fig**). KD of *Nacα-RNAi* resulted in reduced ABD-A protein staining in pupal hearts (**S5B Fig**), opposite of what we see with ABD-B protein levels (**Fig 3E**). Combined *Nacα-RNAi* with *abd-A* overexpression led to lethality at pupal stages. These data demonstrate the specificity of the cardiac rescue with the *Hox* gene *Abd-B* but not *abd-A*. Lastly, co-expression of *Nacα-RNAi* with an inhibitor of apoptosis (Death-associated inhibitor of apoptosis 1, DIAP1) did not rescue the heart (**S6 Fig**), suggesting that the *Nacα*-RNAi mediated heart loss cannot simply be prevented by inhibition of canonical cell-death pathways.

We also tested whether *Abd-B* KD could rescue the cardiac function defects produced by less robust KD of *Nacα*, using the *tinHE*-GAL4 driver, that drives moderate expression in cardiomyocytes during embryonic and pupal development and in the adult. *Nacα* KD with *tinHE*-GAL4 caused cardiac constriction but no effects on heart period or intervals (**S7A and S7B Fig**). KD of *Abd-B* using *tinHE*-GAL4 did not produce significant changes in heart function compared to controls, except for a prolonged systolic interval. Combining *Nacα-RNAi* with *Abd-B-RNAi* reversed the constricted diastolic and systolic diameter phenotype of *Nacα-RNAi;UAS-Stinger::GFP*, such that the phenotype mirrored controls or *Abd-B-RNAi;UAS-Stinger KD*. An improvement in circumferential myofibrillar organization is also evident with co-KD of *Nacα and Abd-B* (**S7C Fig**). These data suggest that *Abd-B and Nacα* co-KD restored heart formation with the strong *Hand4.2* driver, and also normal heart function and myofibrillar organization with the weaker *tinHE*-GAL4 driver. This means that functional maturation requires *Nacα* to restrict *Abd-B* function to the posterior of the heart.

## *Nacα* is required in the embryo to pre-program cardioblasts for appropriate cardiac remodeling

Prior to pupal cardiac remodeling, a larval heart was still present despite *Nacα* KD using the strong cardiac driver *Hand4.2*-GAL4 (**Figs 1E and 3E**). However, these larval heart tubes were thinner and the cytoskeletal structures less prominent than controls (**Figs 3E and** S1B), reminiscent to the constricted adult heart phenotypes with the weaker driver *tinHE*-Gal4 (**Figs 2 and** S7). These observations suggest that *Nacα* may have earlier developmental functions in addition to a role in metamorphosis. We therefore wanted to temporally dissect *Nacα's* function in the heart by knocking down its expression during different developmental stages. We generated a *Hand4.2*-GAL4 driver line that included two copies of a temperature-sensitive allele of GAL80 driven by a ubiquitous promoter (*tubulin*-GAL80$^{ts}$), which we termed HTT [9]. At the permissive temperature (18˚C), the GAL80 transcriptional repressor prevents GAL4 activation of UAS sites thereby inhibiting transcription of downstream constructs [40]. This temperature-sensitive form of GAL80 protein is unstable at higher temperatures (28–29˚C), thus permitting GAL4 activity at higher ambient temperatures.

Maintaining HTT flies crossed to *Nacα-RNAi* or controls at 18˚C throughout development resulted in normal heart structure (**Fig 4A**) and produced no differences in diastolic diameter,

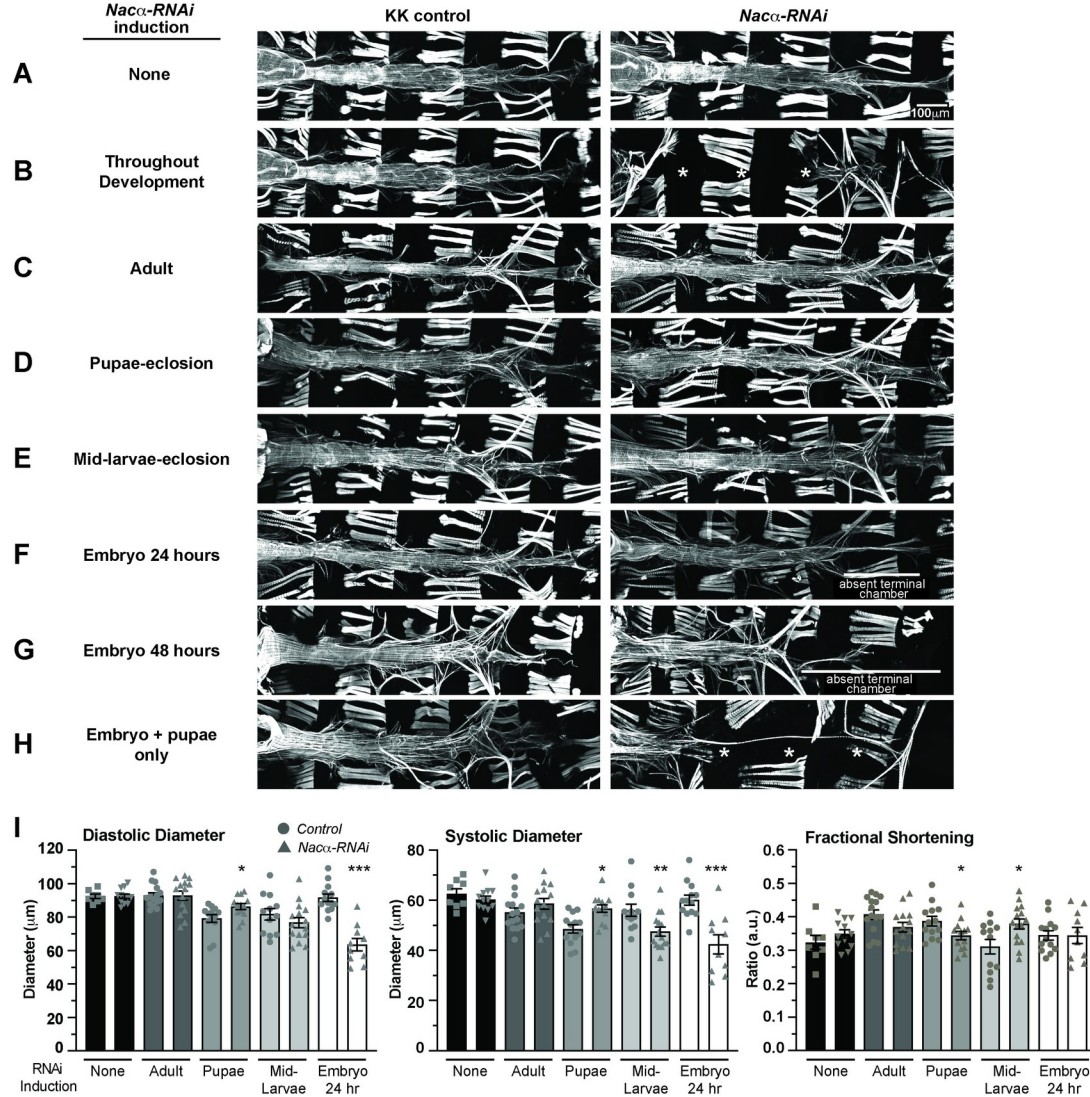

**Fig 4. Temporal regulation of *Nacα*-RNAi expression in the heart.** Using a temperature inducible driver specifically in the heart (HTT, *Hand4.2*-GAL4, *tubulin*-GAL80ts; *tubulin*-GAL80ts), *Nacα*-RNAi was expressed during specific stages of development by controlling ambient temperature to determine its contribution to cardiogenesis. Controls lacking RNAi are kept in similar temperature conditions to account for any developmental effects of temperature on the heart. **A-H.** Phalloidin staining to visualize cytoskeletal structural effects of *Nacα* knockdown (KD). **A,** As a test of GAL80 control of transcription, flies held at 18°C throughout development did not produce changes to heart structure indicating an inhibition of *Nacα*-RNAi transcription. **B,** Exposing flies to high temperatures (28°C) throughout development produced a no heart phenotype similar to the effects of driving *Nacα*-RNAi using *Hand4.2*-GAL4 alone, suggesting an induction of *Nacα*-RNAi transcription and subsequent *Nacα* KD with exposure to higher temperatures. * indicates absence of the heart. **C,** Exposing flies to high temperature during adulthood only for 1 week, **D,** pupae to eclosion, or **E,** mid-larvae to eclosion did not produce gross structural defects in the heart. **F,** Exposing embryos to high temperatures starting at egg-lay up until 24 hours resulted in the absence of the terminal chamber, indicated by white bar. Only thin alary muscles were present. **G,** Extending the high temperature exposure to 48 hours led to similar loss of the posterior heart, indicated by white bar. **H,** Only when the hearts were exposed to higher temperatures during embryonic stage (24 hours) and pupal stage until eclosion (~3 days), were we able to recapitulate the no heart phenotype produced by exposing the heart to constant high temperatures. **I,** Functional analysis of the adult heart following *Nacα* KD at various developmental stages. Maintaining flies at 18°C throughout development or exposure of adult flies to high temperature for 1 week led to no changes in diastolic diameter, systolic diameter, or fractional shortening. High temperature exposure from pupae to eclosion or from mid-larvae to eclosion led to subtle changes in diameters and fractional shortening. Exposure of embryos to high temperatures for 24hr led to constricted diastolic and systolic diameters. * $p<0.05$, ** $p<0.01$, *** $p<0.001$.

systolic diameter or fractional shortening compared to controls (**Fig 4I**), effectively demonstrating GAL80's ability to suppress Nacα-RNAi transcription at 18˚C temperatures. Constant exposure to 28˚C throughout development phenocopies the absence of the heart in adults produced by Hand4.2-GAL4 driver (**Fig 4B**). KD of *Nacα* in adults only for one week by exposure to high temperatures led to largely normal heart structure and function compared to controls (**Fig 4C and 4I**), suggesting that *Nacα* is primarily required developmentally for establishing a normal heart in adults, rather than maintaining its function or structure with age. Remarkably, although lifelong cardiac *Nacα* KD using Hand4.2-GAL4 led to histolysis of the heart during metamorphosis, KD of *Nacα* during pupation only (~3 days at 28˚C) did not produce gross heart defects (**Fig 4D**). The diastolic and systolic diameters were slightly increased, which caused some reduction in fractional shortening (**Fig 4I**). Even when we induced *Nacα* KD earlier, starting at mid-larval stages through metamorphosis until eclosion (~5 days at 28˚C), a period with substantial developmental growth requiring high levels of protein translation, heart structure was unaffected (**Fig 4E**). This lack of phenotype is remarkable, as disruption of NAC is associated with proteostasis and ER stress, which often leads to cell death [41–43]. These results suggest that KD of *Nacα* in the developing heart, even for relatively long durations does not unequivocally lead to cell death. When *Nacα* was knocked down during embryonic stages only (egg-lay up to 24 hours) and subsequently reared at 18˚C until dissection to prevent *Nacα* KD at later stages, most adult hearts remained intact, but considerably constricted with smaller diastolic and systolic diameters (**Fig 4F and 4I**). Interestingly, the posterior terminal heart chamber was absent in most cases (**Fig 4F**), which suggests that the no-heart phenotype observed with continuous KD likely arises from developmental defects already in the embryo. Extending exposure to higher temperature during embryonic stages until 48 hours after egg lay, produced similar phenotypes compared to 24hr exposure, with the presence of a heart tube but without a terminal chamber (**Fig 4G**). Remarkably, when flies were exposed to higher temperatures during both embryonic stages (24 hours) and pupal stages (3 days; for a total of 4 days), with a return to 18˚C during larval development, an almost complete 'no-heart' phenotype was reproduced in most cases (**Fig 4H**). This suggests that *Nacα* KD only during combined embryonic and pupal stages produces a nearly complete loss of heart structures, but at either stage alone was insufficient for such a severe phenotype.

We measured *Nacα* mRNA expression in adult hearts of HTT flies subject to pupae only *Nacα* KD (**S8 Fig**). We detected significantly reduced *Nacα* expression shortly after eclosion, suggesting that despite reduced *Nacα* levels, the heart remodels into a largely normal heart (**Fig 4D**). We also measured *Nacα* mRNA levels in early pupal hearts, that were subject to *Nacα* KD only in the embryo (24 hours) before being returned to 18˚C (**S8 Fig**). Interestingly, we also found reduced *Nacα* mRNA levels in early pupae at the onset of metamorphosis, suggesting that embryonic KD of *Nacα* reprograms cells leading to longer term effects such as reduced *Nacα* expression later in development even without RNAi induction. This reduced *Nacα* expression early in pupae leads to changes in cardiac remodeling and constricted adult hearts (**Fig 4F and 4I**). However, this is still insufficient to completely histolyze the heart, unless subject to additional KD during pupal stages. Some extra-cardiac tissues are also collected during heart dissection in pupae and adult, such as few fat and muscle cells where Hand4.2-Gal4 is not expressed, thus the level of *Nacα* KD is likely an underestimation.

These results suggest that *Nacα's* role in driving heart morphogenesis has a temporal component, perhaps regulating several different processes during cardiac development. Furthermore, the observation that *Nacα* KD during embryonic as well as pupal stages is needed to induce histolysis of nearly the entire heart tube during pupal cardiac remodeling suggests that *Nacα* plays an essential role in embryonic heart development by programing cardiac cell fate. This embryonic requirement seems to be partially compensated for by sufficient *Nacα*

function during heart remodeling. However, additional reduction of pupal *Nacα* exacerbates the changes in embryonic cardioblast programming, together leading to a failure of the larval heart to respond to remodeling cues during metamorphosis, thus causing histolysis.

## *Nacα* alters cell-fate in human Multipotent Cardiac Progenitors and is modulated by *Hox* genes

Since our results suggest that *Nacα* plays a role in establishing cell identity and cell-fate of cardiac cells in *Drosophila*, we wanted to determine whether a similar role could be observed in other model systems, such as cardiomyocytes derived from human iPSCs (hiPSCs). We subjected human iPSC-derived Multipotent Cardiac Progenitors (MCPs) [44] to siRNAs against the human ortholog of *Nacα*, *NACA*, to assess their effects on cell proliferation and differentiation (**Fig 5**) [9]. We evaluated the propensity for MCPs to spontaneously differentiate into different cell types and calculated their relative proportions, by staining with *α-Actinin1* (*ACTN1*) for cardiomyocytes (CM), *Transgelin* (*TAGLN*) for fibroblasts, and *Cadherin 5* (*CDH5*) for endothelial cells. Total cell number was quantified by counting the number of DAPI-positive nuclei. The proportion of differentiated cell types was assessed nine days after siRNA treatment, a timepoint when active KD is no longer expected [9].

Treatment of MCPs with *NACA siRNA* did not change total cell count compared to controls (**Fig 5A and 5B**), but significantly decreased the proportion of CMs (**Fig 5C and 5D**), and increased the proportion of fibroblast (**Fig 5E and 5F**). The proportion of endothelial cells remained unchanged (**S9A and S9B Fig**). These results suggest that *NACA* may play a role in directing cell fate toward a cardiac program and its absence shifts these fates toward fibroblast differentiation.

We wondered whether the effect of *NACA* on CM differentiation can be similarly reversed by human orthologs of *Drosophila Abd-B*. Therefore, we tested the effect of siRNAs against human *Homeobox C12* (*HOXC12*) or *Homeobox D12* (*HOXD12*), which have close sequence homology with *Abd-B*. Treatment of either *HOXC12 and HOXD12 siRNAs* individually did not have a significant effect on total (**Fig 5A**), cardiomyocyte (**Fig 5C**), fibroblast (**Fig 5E**)**,** or endothelial (**S9A Fig**) cell populations compared to controls. When we tested combinations of *NACA* siRNA with *HOXC12* and *HOXD12* siRNA, or all three, we found that the *Hox* genes could alter *NACA* phenotypes. While treatment with *NACA* siRNA led to a reduced trend in total cell populations, combined KD of *NACA* and *HOXC12* and *HOXD12* led to total cell populations that were significantly higher than *NACA siRNA* treatment alone and were more similar to controls (**Fig 5A and 5B**). More striking is the reversal in the proportion of cardiomyocyte and fibroblast populations when *NACA siRNA* was combined with *HOXC12* and *HOXD12*. Co-transfection of *NACA* with the selected *Hox* genes led to increased CM (**Fig 5C and 5D**) and reduced fibroblasts (**Fig 5E and 5F**) such that they are significantly different compared to *NACA* siRNA treatment alone and no longer different compared to controls. Efficiency of siRNA KD following transfection is expected to be maintained even when siRNAs against 3 different gene targets are combined, as demonstrated in previous experiments [7, 9]. These results suggest that the reduction in CM and increase in fibroblast caused by *NACA* KD is dramatically rescued upon *Hox* co-KD, thus restoring CM differentiation of these pluripotent cells (**Fig 5G**). It remains possible that *NACA* interacts with other *Hox* gene*s* not tested here, in mammalian cardiac tissue to direct morphogenesis. Furthermore, overexpression of *Hox* genes alone would be expected to have an effect on the ratio of cardiomyocyte and fibroblasts and their interaction with *NACA* could also be explored. These results are consistent with observed cardiac differentiation and genetic interactions in *Drosophila* and that *Nacα/NACA* activity in the heart may in part be mediated by posterior *Hox* genes.

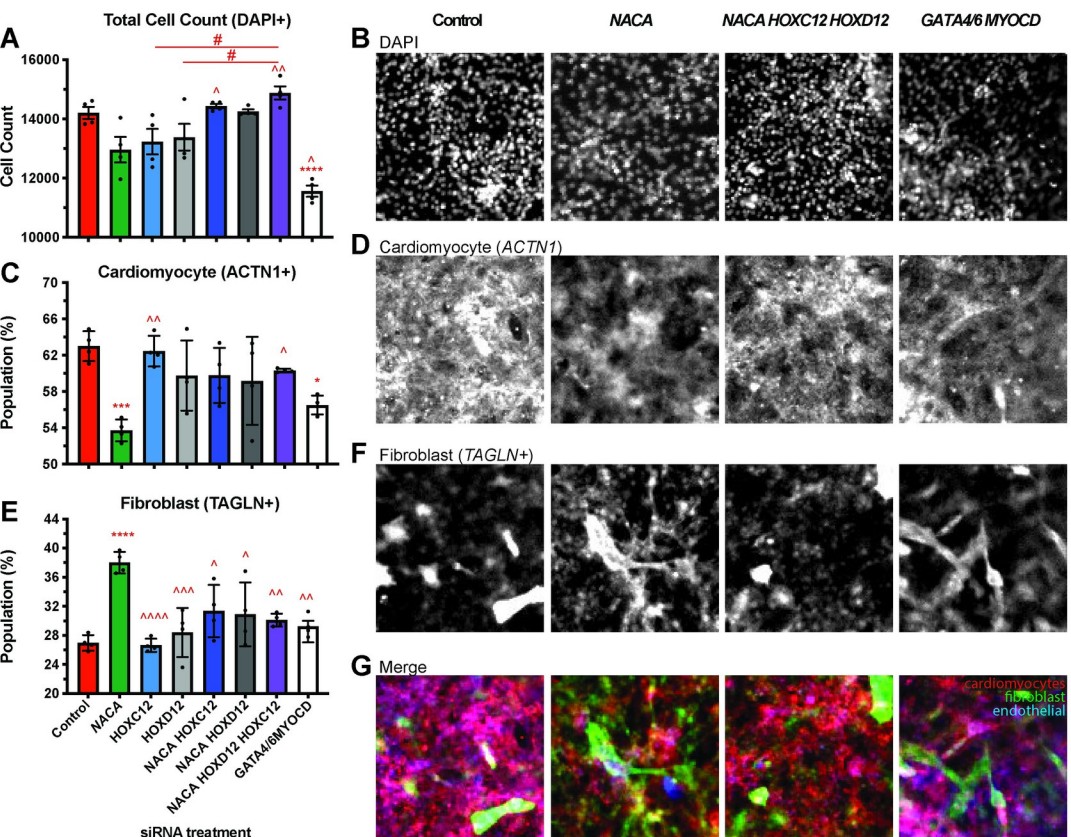

**Fig 5. *Nacα* and *Hox* genes interact to redirect differentiation of Multipotent Cardiac Progenitors (MCPs). A,C,E,** Quantitation of differentiated cell populations 9 days after siRNA treatment. **B,D,F.** Representative images of immunohistological staining for select conditions. **A,B,** Total Cell Populations following siRNA treatment were not significantly changed compared to controls, except for *Gata4/6,MyoCD* siRNA condition which reduced overall cell count. Knockdown (KD) of *Nacα* (green), *HOXC12* (light blue), *HOXD12* (light gray) singly resulted in cell populations that trended lower. This decrease was reversed and significantly different upon combined transfection of *Nacα siRNAs* with *Hox* genes, compared to single siRNA transfections. **C,D,** *Nacα* KD alone significantly decreased the proportion of cardiomyocytes (ACTN1+) compared to controls, while treatment with HOXC12 or HOXD12 siRNA individually, had no effect. Combined transfection of *Nacα* and *Hox* genes reversed the decrease in cardiomyocyte population and was significantly different compared to *Nacα* KD alone and no longer different compared to controls. *Gata4/6,MyoCD* KD also significantly lowered the proportion of cardiomyocyte populations. **E,F,** *Nacα* KD increased the proportion of fibroblasts (TAGLN+) compared to controls. Combined KD of *Nacα* with any of the *Hox* genes did not alter fibroblasts numbers compared to controls but were significantly reduced compared to *Nacα* KD alone. **G,** Images of merged staining of cardiomyocyte, fibroblast, and endothelial cells shows the decrease in cardiomyocyte (red) and increase in fibroblast (green) staining when *NACA* is knocked down compared to controls. This is reversed upon co-KD with *HOXC12 and HOXD12*. Significance * vs. control. ^ vs. *Nacα*. # comparison is indicated by line. * $p<0.05$, ** $p<0.01$, *** $p<0.001$, **** $p<0.0001$.

As a positive control and a means of comparison, we transfected MCPs with siRNAs against cardiogenic transcription factors GATA Binding Protein 4 and 6 (*GATA4/6*) and Myocardin (*MYOCD*) [45, 46]. *GATA4/6,MYOCD* KD caused a significant decrease in total cell number (**Fig 5A and 5B**) and proportion of cardiomyocytes (**Fig 5C and 5D**), no change in the proportion of fibroblast (**Fig 5E and 5F**), and an increase in the proportion of endothelial cells (**S9A and S9B Fig**). Thus, the effect of *NACA* KD on the proportion of cell types were different compared to the cardiogenic factors. The only similarity observed between *NACA* and *GATA4/6, MYOCD* KD was a decrease in the proportion of cardiomyocytes, although *NACA* produced a greater decrease (**Fig 5C and 5D**). These results suggest that *NACA KD* induced alternate cell fates, possibly by a different mechanism to that of these cardiogenic transcription factors.

## Knockdown of individual SRP subunits cause distinct cardiac phenotypes

NAC is just one of the protein biogenesis quality control mechanisms that is found at the ribosomal exit site sifting through emerging nascent polypeptides and guiding protein fates. SRP is another protein complex involved in controlling protein biogenesis by binding a disparate subset of nascent polypeptides destined for the ER, wherein its targeting is influenced by the activity of NAC. Therefore, we wanted to examine if disruption of SRP function in fly hearts would result in similar defects compared to *Nacα* KD. Eukaryotic SRP is composed of a 7SL SRP RNA that holds the conformation of 6 proteins (SRP9, SRP14, SRP19, SRP54, SRP68, and SRP72) and targets approximately 30% of newly synthesized proteins to the ER (**Fig 6A**) [47]. SRP primarily recognizes the N-terminal hydrophobic sequences of emerging nascent polypeptides, but has also been shown to bind nascent chains even when target sequences are not yet accessible [48]. In yeast, SRP also binds nascent chains with internal transmembrane domains [49]. Once bound, SRP arrests translation of the nascent chain until the SRP-ribosome complex binds with the SRP-receptor (SR) anchored to the ER, where translation of the nascent chain is restarted and co-translationally released through the Sec61p translocase for insertion.

Each subunit of the SRP complex exhibits specialized roles in the binding and translocation process (**Fig 6A**) and therefore, each subunit could confer specialization of SRP function through recruitment of cofactors or selective targeting of nascent proteins. We therefore knocked down each of the SRP subunits individually, as well as the β-subunit of SRP Receptor (*SRPR-β*), to explore their role in heart development and how their phenotypes compare to *NACα* and *bic* KD. We used two different RNAi lines when available and showed examples of RNAi lines with more severe phenotypes (**S1 Table**). RNAi mediated KD of either *Srp9* or *Srp14* subunits in the heart driven by *Hand4.2*-GAL4 using two different RNAi lines did not produce gross differences in the heart structure compared to controls (**Fig 6B–6D**). In contrast, KD of *Srp68* led to complete absence of the adult heart, similar to *Nacα* KD (**Fig 6E**). Upon KD of *Srp72*, the heart tube was still present which is unlike the *Nacα* KD phenotype. The very anterior segment, called the conical chamber, remained very constricted, reminiscent of its larval structure (**Fig 6G and 6H**). This larval aorta-like structure suggests that this section of the heart failed to undergo remodeling during metamorphosis. KD of *Srp19* led to a partial heart phenotype, where the anterior segment of the adult heart tube is absent, but retained a posterior segment, including the terminal chamber (**Fig 6F**), again unlike what was observed with *Nacα* KD. KD of *Srp54*, which recognizes and binds the signal sequence on the nascent polypeptide, led to a malformed heart with missing cardiomyocytes in random positions throughout the tube (**Fig 6I and 6J**). KD of *SR-β* led to a heart tube with missing cardiomyocytes, which were most often the internal valves of the heart (**Fig 6K–6M**). Interestingly, *in vivo* imaging of fluorescently labeled hearts in early pupae demonstrated that an intact larval heart tube forms with KD of any SRP components (**S10 Fig**), suggesting that their requirement prior to pupal remodeling of the heart may be less critical. Considering *Srp19* and *Srp68* KD largely abolished adult hearts, the presence of an intact heart tube prior to cardiac remodeling in pupae suggests, that like *NACα*, their activity is critically required during cardiac remodeling and morphogenesis.

In summary, the KD of individual NAC and SRP subunits led to a range of distinct cardiac phenotypes that suggests each subunit and complex may function with distinct cell-type, regional and temporal specificities during cardiac development. Further characterization of SRP phenotypes in the embryo and during metamorphosis, measuring the magnitude of knockdown of each SRP subunit, and proteomic analysis would help decipher whether each subunit contributes specialized function in development and interacts with other cofactors that could target subsets of genes differently between SRP and *NACα*.

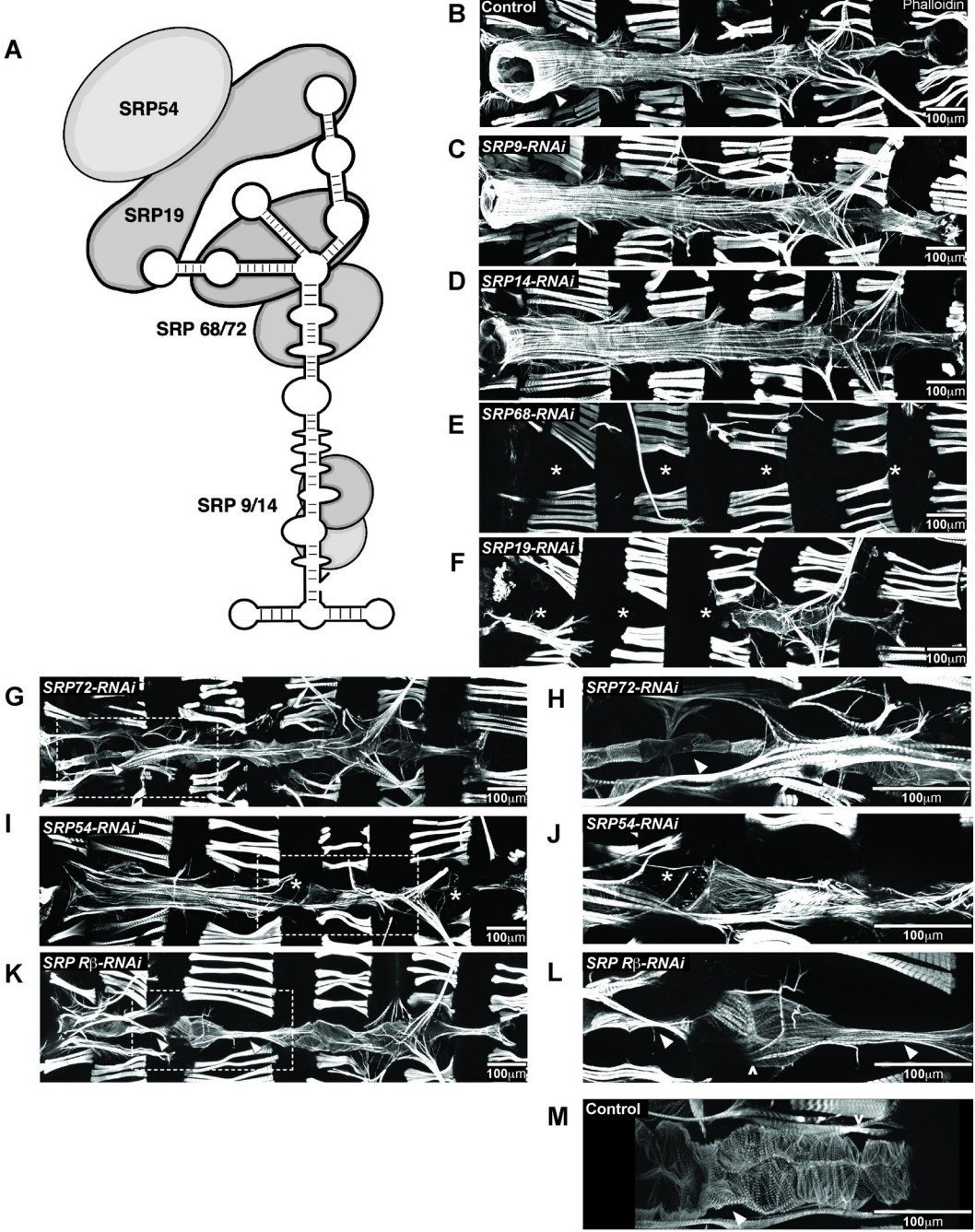

**Fig 6. Knockdown (KD) of SRP subunits in the *Drosophila* heart caused distinct heart defects. A**, The Signal Recognition Particle (SRP) is composed of an RNA molecule holding together 6 SRP subunits. **B-M,** Individual SRP subunits were KD by RNAi using the heart specific driver *Hand4.2*-GAL4 and adult flies were stained with phalloidin to determine their contribution to heart structure and development. * indicates missing heart segments or cardiomyocytes. KD of **C**, SRP9 and **D,** SRP14 subunits, responsible for elongation arrest during translation, did not lead to gross alterations in heart structure and were comparable to **B,** controls. Arrowhead in control image point to the conical chamber. **E**, KD of SRP68 led to complete loss of the heart. **G,** Interestingly, KD of SRP72, a binding partner to SRP68 led to the presence of a heart tube. **H,** Higher magnification of the conical chamber, marked by an arrowhead, showed that the conical chamber was constricted compared to controls (**B**), and resembled a larval aorta. **F**, KD of SRP19, led to a partial heart phenotype, where the anterior region of the heart was absent but the posterior end remained. **I**, KD of SRP54 led to a heart tube but with missing heart cells (indicated by asterisks) in random regions of the heart. **J**, Higher magnification of missing cardiomyocyte. **K**, KD of the SRP Receptor-β, a subunit of the receptor anchored to the ER membrane that binds to the SRP-ribosome-nascent chain complex, led to segments of the heart, usually the valves, that were constricted and larval like, indicated by the arrowhead. **L**, Higher

magnification of the narrowed heart tube. Ostia structures are still present as marked by ^. **M,** As comparison, valve cells in controls (marked by arrowhead) are wider than SRPβ KD and are densely packed with myofibrils.

## Discussion

Nascent polypeptide Associated Complex (NAC) and the Signal Recognition Particle (SRP) are integral to protein biogenesis. Our work suggests that they are pertinent for establishing distinct proteomic landscapes that shape cell identity and direct developmental fates (**Fig 7**). We demonstrated a developmental role for the NAC subunit *Nacα*, in the *Drosophila* heart (**Figs 1 and** 2) and Human Multipotent Cardiac Progenitors (MCPs; **Fig 5**) that influenced cardiac cell identity and morphogenesis through associations with *Hox* genes (**Fig 3**). These cardiac defects were triggered by knockdown (KD) of *Nacα* at specific stages in development (**Fig 4**). Similarly, disruptions in SRP subunit expression led to cardiac cell type specific defects in the fly with some changes in heart morphology occurring during later, critical stages of development (**Fig 6**). These results suggest that *Nacα* function and interaction with select protein targets were temporally regulated and that SRP subunits could function similarly. Components involved in protein translation are a growing family of genes associated with tissue-specific and developmental defects [50], including the heart [51, 52]. For example, in mice, mutants of the ribosomal protein *Rpl38* displayed specific axial skeletal patterning defects attributable to reduced protein levels of a subset of *Hox* genes (HOXA5, HOXA11 HOXB13) without changes in global protein translation [53, 54]. Patients with Diamond Blackfan Anemia (DBA), a bone marrow failure syndrome caused by ribosomopathies, exhibit higher manifestations of CHD compared to the general population [55]. In other cases, ribosomopathies led to CHD defects without overt hematological abnormalities [9, 56, 57]. *Nacα* specifically has been associated with Tetralogy of Fallot [11] and increased myocardial mass [10]. Within the SRP pathway, *SRP54* mutations cause Schwachman-Diamond-like syndrome, a condition associated with CHDs [58, 59]. Furthermore, within the Pediatric Cardiac Genomic

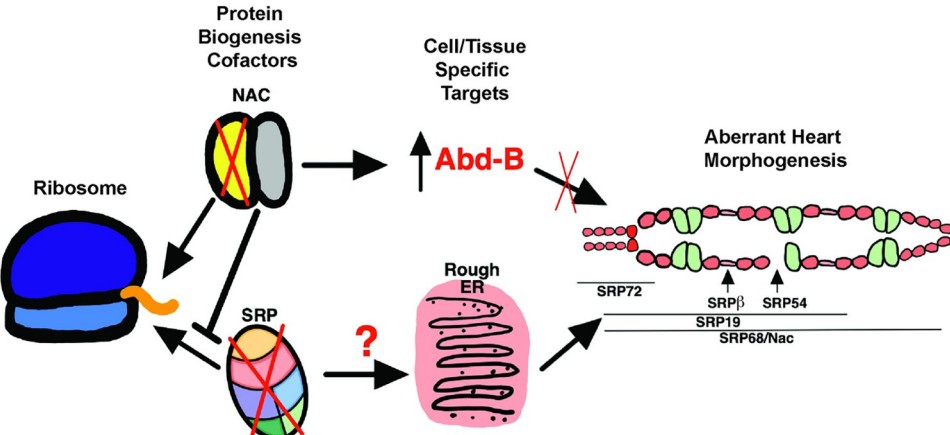

**Fig 7. A schematic diagram of protein translation at the ribosome exit site.** Nascent polypeptide cofactors NAC and SRP select for gene targets, regulating their expression in a tissue and/or cell specific manner to regulate heart morphogenesis. Our work suggests that the posterior determining *Hox* gene *Abd-B* is a target of *Nacα*, as the knockdown of *Nacα* led to *Abd-B* misexpression and disruption of heart morphogenesis through failure of cardiac remodeling during metamorphosis. Knockdown of SRP subunits each led to distinct heart defects: SRP72 knockdown disrupted conical chamber morphogenesis, SRP-Rβ targeted internal valve cells, SRP54 led to missing cardiomyocytes, SRP19 led to loss of anterior segment of the heart, while SRP68 led to a no heart phenotype similar to *Nacα* knockdown. Future work can help uncover the nature of the distinct phenotypes whether due to dissimilar KD levels or targeting of unique transcripts by individual SRP subunits to direct different cell fates.

Consortium cohort [60], the SRP Receptor-α was associated with atrial septal defects while SRP Receptor-β was associated with cyanotic congenital heart disease including ventricular septal defect. As genomic studies of patients uncover new variants associated with CHD, a closer examination of variants within translational genes and their co-occurrence with gene variants within developmental genetic networks is warranted.

It is remarkable that the complete histolysis of the adult fly heart observed during pupal stages required KD of *Nacα* during both embryonic and pupal stages, either one alone was not sufficient (**Fig 4D–4H**). This was surprising, since during pupal stages the heart undergoes extensive remodeling, requiring significant protein biogenesis. Yet, pupal-only KD (or mid-larval to eclosion KD) of *Nacα* did not induce cell death, nor cause significant changes in heart structure or function. This suggests that *Nacα* function during embryonic stages may create a proteomic landscape establishing long-term programming of cardiomyocyte identity properly poised for future developmental responses, such as remodeling to the adult heart during metamorphosis. An embryonic requirement for later cardiac remodeling and survival was also found for ribosomal protein gene *RpL13* [9]. However, in the case for *RpL13*, KD in the embryo alone resulted in significant heart loss during later developmental stages. Thus, again translational perturbances in the embryo can have long-lasting effects, suggesting a mechanism by which preprograming in the embryonic heart is critical for appropriate cardiac remodeling later in development. How translational proteins and mechanisms contribute to redefining cell identity would need to be better understood.

Our studies found that KD of *Nacα* restricted by *Hand4.2*-GAL4 to mid- to late stages of embryo development (beginning stage 14 until hatching) specifically in cardiac relevant cells induced posterior terminal heart chamber defects that were evident in adult morphology (**Fig 4E and 4F**). This suggests spatial specificity in *Nacα* function with embryonic origins targeting posterior cardiac cells more severely compared to more anterior regions. We also demonstrated that *Nacα* genetically interacts with a posteriorly expressed *Hox* gene, *Abd-B*, later during pupal stages, which further supports a stronger requirement for *Nacα* in posterior heart patterning (**Fig 3**). The KD of *Nacα* in cardiac cells permits ABD-B protein expression in the anterior regions of the larval heart and aorta, where it is normally absent (**Fig 3C–3E**). Ectopic *ABD-B* expression could occur through direct disinhibition of *Nacα* translational control of *Abd-B* mRNA. Alternatively, *Nacα* KD could have an indirect effect on cell identity that aberrantly turns on *Abd-B* transcription and subsequent protein expression. The regional specificity in *Nacα* activity is consistent with a previously described role for NAC in the posterior patterning of the developing *Drosophila* embryo [19, 20]. In the embryo, posterior patterning proteins, such as OSKAR and NANOS, accumulate in the posterior end of the embryo. Components of the NAC complex were necessary to restrict mRNA localization of OSKAR and NANOS to the posterior end and spatially restricting their subsequent translation [19, 20]. Absence of NAC components resulted in ectopic expression of posterior patterning proteins anteriorly leading to a lethal, bicaudal phenotype. Future studies will be aimed at determining the mechanisms that regulate *Nacα* activity that drives these anterior-posterior specific responses in the heart, including the regulation of *Abd-B* expression and activity. The failed rescue by the more anteriorly expressed *Hox* gene *abd-A* as well as the low protein sequence homology outside of the highly conserved homeodomain amongst *Hox* protein is consistent with a specific interaction between *Nacα* and *Abd-B*. *Hox* genes play a crucial role in early cardiac specification, patterning and remodeling [61] and *Nacα* translational regulation could be a novel mechanism of regulating *Hox* genes as well as other cardiac relevant targets that may contribute to pathogenesis of CHD.

Our work using MCPs demonstrated a role for *Nacα* in human cardiac cell specification and differentiation. KD of *Nacα* redirected progenitor cell fates from a cardiomyocyte lineage

toward a fibroblast lineage (**Fig 5**), a population profile that was distinct from KD of cardiogenic transcription factors *Gata4/6* and *MyoCD* (**Fig 5**) [62, 63]. This suggests that *Nacα* utilizes distinct mechanisms for refinement of cardiac cell fates, but it remains to be determined how these processes are coordinated. Interactions between *Hox genes* and *Nacα* were also consistent between flies and mammalian cells and thus have implications for *Hox gene* mediated anterior-posterior patterning crucial for proper morphogenesis in mammals. Complex signaling events coordinate patterning of cardiac progenitor populations within subdivided domains defined by unique transcriptional programs contributing to the development of distinct heart structures [64]. Transcriptional profiles within these subdomains have implicated specific *Hox* gene expression (*i.e. Hoxa1* and *Hoxb1*) to establish cellular identity and cell fate [64, 65]. Future work in mammals can determine whether NAC and SRP activity contribute to targeted expression of *Hox* genes within distinct subdomains, as suggested by our work in flies and human MCPs.

Protein biogenesis and its regulation at the ribosome exit site relies on a network of nascent protein chaperones. Because of the interactions between NAC and SRP that refine their ability to target specific nascent polypeptides emerging from ribosomes, we explored the effects of disrupting SRP activity on fly heart development and compared them to *Nacα* KD. KD of individual SRP subunits led to unique and varied cardiac phenotypes (**Fig 6**). Furthermore, SRP's influence on cardiogenesis is temporally regulated, as gross changes in morphology occur during crucial developmental stages, such as the disappearance of the heart during metamorphosis induced by *Srp68* KD (**Figs 6E and** S6). Further characterization of SRP function in heart morphogenesis will be focused on whether individual SRP subunits contribute to cell-fate specific translational activity through interactions with differentially expressed cofactors. This would help clarify the distinct cardiac phenotypes induced by KD of each subunit and uncover novel aspects of protein biogenesis control in a cell-type specific manner. In contrast to *Nacα*, the SRP complex targets nascent proteins destined for insertion into the rough ER and subsequent translocation to the plasma membrane or for secretion. Therefore, SRP targets are likely enriched for membrane receptors. Thus, interruption of receptor function and non-cell-autonomous signaling may be partially involved in the cardiac phenotypes caused by disruption of SRP activity. For example, cardiac cells may respond aberrantly to ecdysone steroid hormone secreted by the ring gland during metamorphosis due to a lack of Ecdysone Receptor expression at the cardiac cell membrane. Indeed, this may in part be a contributing mechanism, as *Srp72* KD in the heart resulted in constriction of just the conical chamber in the anterior region of the adult heart, which resembled an unremodeled larval heart (**Fig 6G and 6H**), as previously observed for inhibition of Ecdysone Receptor function [38, 66]. Still, why there appears to be regional specificity to this lack of Ecdysone Receptor response, despite *Srp72* KD throughout the heart, remains to be determined.

In summary, our work demonstrated specific roles for the NAC and the SRP complex in cardiac development, using *Drosophila* and human MCPs, and offers novel mechanisms in the regulation of the cardiac proteome to establish cell identity and tissue patterning (**Fig 7**). How generic components of the translational machinery could lead to targeted effects on heart development and function is likely layered in a complex process that remains to be explored.

## Materials and methods

### Drosophila strains

Heart-specific control of transcription was achieved by the GAL4-UAS system [67], in which the following cardiac relevant GAL4 drivers were used: Hand4.2-GAL4 [33, 68, 69], *tinCΔ4--*GAL4 [70], *tinD*-GAL4 [71], and *tinHE*-GAL4 [72]. *Dot*-GAL4 driver line [34] was used to

express in pericardial cells. *Drosophila* GD and KK RNAi collection lines along with appropriate controls were obtained from the Vienna Drosophila Resource Center (VDRC) [73]. KK and GD controls are the original background fly line wherein the RNAi construct was inserted. Two copies of a temperature-sensitive *tubulin*-GAL80 (*tubulin*-GAL80$^{ts}$) were recombined and combined with the Hand4.2-GAL4 driver (Hand4.2-GAL4, Tubulin-GAL80$^{ts}$; Tubulin-GAL80$^{ts}$, HTT; [9]). All fly lines are listed in **S1 Table**.

## Developmental regulation of gene expression

The HTT line allows for transcriptional control of constructs fused to a UAS enhancer by manipulation of ambient temperature. At 18˚C, GAL80 is intact and blocks GAL4 mediated activation of the UAS enhancer. A shift to 29˚C destabilizes GAL80 protein, permitting GAL4 binding to the UAS enhancer to induce transcription. As control, flies were placed in 18˚C or 28˚C throughout development until eclosion when an intact or an absent heart were expected, respectively. For *Nacα* knockdown (KD) in embryonic stages, female and male flies were placed in 29˚C and embryos collected every 2 hours. Embryos were maintained in 29˚C for 24 hours and then shifted to 18˚C for the remainder of development. Another set of embryos were maintained in 29˚C for 48 hours before shifting to 18˚C. For *Nacα* KD during pupal stages, fly crosses were maintained in 18˚C until wandering 3$^{rd}$ instar larvae were collected. They were then immediately placed in 29˚C until eclosion. For KD of *Nacα* starting at mid-larvae, embryos from fly crosses at 18˚C were collected every 2 hours, maintained in 18˚C for 16 days when larvae were approximately at L2 stages. Larvae were then placed in 29˚C until eclosion. Finally, to test *Nacα* KD during embryonic and pupal stages only, embryos were collected from fly crosses at 29˚C for 24 hours as described above, and then returned to 18˚C to develop through larval stages. Wandering larvae were then collected and then placed in 29˚C until eclosion. Adult flies were assessed for heart function using SOHA (see below) and immunostained to examine structure (see **Fig 4**).

## Immunostaining of adult drosophila hearts

Adult flies were dissected and treated with 10mM EGTA in PBT (PBS + Triton-X-100; 0.03% Triton X-100), for 2 minutes to maintain a relaxed state of the heart. Hearts were then fixed with 4% PFA in PBT for 20 minutes, followed by three 10-minute PBT washes. Hearts were stained with primary antibodies (EC11 *Pericardin* and *Abd-B* from Developmental Studies Hybridoma Bank, DSHB; *abd-A* antibody from Santa Cruz Biotechnology) and incubated overnight in 4˚C. Hearts were then washed with PBT three times for 15 minutes each, followed by incubation with fluorescent secondary antibodies (1:500, Jackson ImmunoResearch Laboratories, Inc.) and Alexa Fluor conjugated phalloidin (1:300, Life Technologies) in 4˚C overnight. Hearts were then washed with PBT three times for 15 minutes each and then once with PBS. Hearts were mounted using ProLong Gold Mountant with DAPI (Life Technologies). Immunostained preparations were visualized with an Imager.Z1 equipped with an Apotome2 (Carl Zeiss, Jena), Hammamatsu Orca Flash4.0 camera, and ZEN imaging software (Carl Zeiss).

## Live-imaging of the heart in drosophila pupae

For fluorescence-based heart function analysis, we crossed a fly line that contained both Hand4.2-GAL4 and a heart enhancer fused to tandem-Tomato fluorescent protein (tdtK), to controls, *Nacα*-RNAi, or *bicaudal*-RNAi flies [73]. White pupae (WP) from these crosses were collected and lined up on small petridishes with dorsal side facing up. Dishes were placed on a Zeiss Imager M1 equipped with a Hamamatsu Orca-Flash 4.0 Digital Camera (C11440), and

heart images were captured every 2 minutes, for a duration of about 80 hours at room temperature (fluctuations of approx. 18–23˚C) using ZEN imaging software (Carl Zeiss). Movies were formatted and compiled using ImageJ and iMovie (Apple).

## Heart function analysis

Assessment of *Drosophila* heart function and structure using the Semi-automatic Optical Heartbeat Analysis (SOHA) method as previously described [74, 75]. Briefly, four-day old adult flies were anaesthetized with FlyNap (Carolina Biological Supply Co, Burlington, NC) and dissected in oxygenated artificial hemolymph to expose the beating heart within the abdomen. Hearts were placed on an Olympus BX61WI microscope while being filmed through a 10x water immersion lens with a high-speed digital camera (Hamamatsu Photonics C9300 digital camera) using HCI image capture software (Hamamatsu). High-speed movies were analyzed using the Semi-automated Optical Heartbeat Analysis (SOHA) software [74, 75]. Parameters measured include Heart Period (HP), Diastolic Interval (DI), Systolic Interval (SI), Arrhythmicity Index (AI), Diastolic Diameter (DD), Systolic Diameter (SD) and Fractional Shortening (FS) (**Fig 2A**). FS, a measure of contractility, is calculated using the following equation FS = (DD-SD)/DD.

## Real-Time quantitative PCR

Approximately 25–30 hearts from early pupae (20-24hr APF) or adults were pooled for each biological sample. Three biological replicates were sampled for each condition. Total RNA was extracted using TRIzol (Invitrogen) and RNA (500 ng) was reverse transcribed to cDNA using QuantiTect Reverse Transcription Kit (Qiagen) and subject to qRT-PCR using the FastStart Essential DNA Green Master reagents (Roche) and LightCycler 96 Instrument (LC96, Roche). The data were analyzed using the $\Delta\Delta$Ct method using Ribosomal Protein 49 (*Rp49*) as a normalization control. Drosophila *Nac$\alpha$* primer sequences: F- AAGGCCAGGAAGATCATGCT; R- ATCCTCGATCTTGGCCTCAC. Rp49 primer sequences: F-AAACGCGGTTCTGCATG AG R-GCCACCAGTCGGATCGATAT.

## MCP cell culture, siRNA transfection, and Immunostaining

A pool of 4 unique siRNA sequences targeting different regions of selected genes and random control were obtained from the human siGENOME library from Dharmacon, Inc. Frozen 5-day old human MCPS [44, 76] were thawed and transfected with siRNAs at 5nM final concentration using Lipofectamine RNAiMAX transfection reagent (Invitrogen). Approximately 20,000 cells were plated in each well of a 384-well plate (Greiner Bio-One) coated with Matrigel Basement Membrane Matrix (Corning). Cells were incubated at 37˚C and media refreshed every second day. Each experiment contained quadruplicate technical replicates per condition and performed on different batches of MCP clones programmed independently.

Immunostaining to identify cell-type composition of cultures were performed 9 days following siRNA transfection. Cells were fixed with warmed 4% paraformaldehyde solution for 30 minutes without agitation, followed by an additional 30 minutes of fixing with agitation. Wells were washed with Phosphate Buffered Saline (PBS) for 10 mins and repeated three times. Samples were then incubated with blocking solution for 30 mins (10% horse serum, 2.5% Triton-X-100, 10% gelatin). Primary antibodies were diluted in blocking solution and added to samples for 1 hour at room temperature (RT): ACTN1 (Sigma, A7811), TAGLN (Abcam, Ab14106), CDH5 (R&D Systems, AF938). Wells were washed with PBS for 15 mins and repeated three times. Samples were incubated with Alexa-conjugated secondary antibodies (Life Technologies) including DAPI diluted in blocking solution for 1 hour at RT. Wells were

washed for 15 minutes and repeated three times, following which samples were imaged using a High-Throughput microscope (ImageXpress, Molecular Devices). Fluorescence was quantified using custom MetaXpress software (Molecular Devices) whereby, the number of total cells and cells positive for ACTN1, TAGLN, and CDH5 in each sample were quantified [44, 76].

### Ethics statement

Sanford Burnham Prebys' Medical Discovery Institute IRB Office's reviewed the use of human derived Multipotent Progenitor Cells and determined that their use in research does not generate identifiable private information or identifiable biospecimens and thus does not involve human subjects.

### Statistical analysis

Statistical analysis was performed using GraphPad Prism (GraphPad Software, La Jolla USA). Data are presented as the mean ± SEM. Human MCP, cardiomyocyte and fibroblast data were analyzed using a one-way ANOVA analysis followed by a Dunnett's multiple comparison post-hoc test. A student's unpaired t-test was used to analyze *Drosophila* heart function data.

### Supporting information

**S1 Fig. Cardiac phenotypes following *Nacα* and *bicaudal* knockdown. A,** Fluorescently tagged hearts using tdtK were imaged at higher magnification in vivo at white pupae stages which displayed intact hearts in control, *Nacα*-RNAi, and *bicaudal*-RNAi expressing hearts, suggesting that the heart histolyzes later, when remodeling during metamorphosis. Arrow heads point to the internal valves that separate the larval aorta from the heart. ^ point to the inflow tracts called ostia. **B,** Pupal dissections at approximately 24-26hr APF stained with phalloidin shows the presence of the fly aorta and heart in controls (left) and with *Nacα* KD (right) at two. magnifications. Arrowheads point to the presence of a heart tube.
(PDF)

**S2 Fig. *Nacα* and *bicaudal* knockdown (KD) using additional RNAi lines.** KD of a second *Nacα* RNAi line (GD v36017) using Hand4.2-GAL4 driver, led to the presence of an adult heart tube. **A,** Hearts were dysfunctional, with significantly increased systolic diameter, that led to severely blunted Fractional Shortening. **B,** Phalloidin staining visualized highly disorganized circumferential fibers (arrowhead). **C,** KD of a second *bicaudal* RNAi line (GD v15453) using Hand4.2-GAL4 led to a no heart phenotype. *-indicates missing heart.
(PDF)

**S3 Fig. Knockdown of *Abd-B* using heart specific driver Hand4.2-GAL4 led to intact hearts with posterior ends (indicated by arrowhead).** Posterior structures were more prominent and dilated compared to controls suggesting incomplete histolysis during cardiac remodeling.
(PDF)

**S4 Fig. Wing blister phenotypes. A,** Wing hearts originate from pericardial cells and are required for wing maturation. In controls, proper wing heart function and hemolymph flow leads to adhesion of dorsal and ventral wing layers. **B**, Normal wing development is also observed with knockdown of *Abd-B*. C, Knockdown of *Nacα* using Hand4.2-GAL4 driver leads to fluid filled wing blisters (*) and crumpled wings (^). D, Co-Knockdown of *Nacα* and *Abd-B* did not rescue the wing blisters.
(PDF)

**S5 Fig. The Hox gene *abd-A* does not rescue the cardiac phenotype induced by *Nacα* knockdown (KD). A**, Still images from video recording of hearts for SOHA analysis. While controls and *abd-A* KD retained a heart structure, Co-KD of *Nacα* and *abd-A* led to a no heart phenotype, indicating an inability for *abd-A* KD to rescue the loss of the heart caused by *Nacα* KD. **B**, *Nacα* KD led to reduced *abd-A* levels in the heart. *-indicates absence of heart structure. ^ indicates the presence of ostia structures.
(PDF)

**S6 Fig. *Nacα* does not interact with *DIAP1*.** Overexpression of an inhibitor of apoptosis (*Diap1*). concurrently with *Nacα* -RNAi does not rescue the loss of the heart. * indicates absence of the adult heart tube.
(PDF)

**S7 Fig. *Nacα* interactions with *Abd-B* to regulate heart function and structure.** Testing interaction of *Nacα* and *Hox* gene *Abd-B* using the cardiac specific *tin*HE-GAL4 driver by **A**, functional, **B**, temporal and **C**, structural assessment. **A**, Knockdown (KD) of *Nacα* (combined with UAS-*Stinger*::GFP to control for UAS binding sites) using *tin*HE-GAL4 caused a decrease in both diastolic and systolic diameters that produced a slight but not significant decrease in fractional shortening. KD of *Abd-B* (combined with UAS- *Stinger*::GFP) did not produce significant. changes in fractional shortening or diameters compared to control but fractional shortening and diastolic diameters were significantly higher compared to *Nacα*;*Stinger* genotype. Combined knockdown of *Nacα* and *Abd-B* produced heart parameters that were not different to controls but recapitulated heart function produced by *Abd-B* KD alone, suggesting that the heart function was rescued. **B**, Temporal parameters were unchanged with *Nacα*-RNAi expression. KD of *Abd-B* lengthened systolic interval compared to controls. Combined *Nacα* and *Abd-B* KD displayed longer systolic intervals similar to *Abd-B* KD alone, suggesting a rescue. **C**, Phalloidin staining of select genotypes. Compared to controls, *Nacα* knockdown disrupted circumferential fiber organization creating gaps in the matrix (similar to Fig 2I). KD of *Abd-B* did not significantly alter circumferential fiber organization. Combined knockdown of *Nacα* and *Abd-B* (2 examples shown) improved circumferential fiber organization compared to *Nacα* knockdown alone. * vs control KKGD. ^ compared to *Nacα*;*Stinger*. * p<0.05, ** p<0.01, *** p<0.001.
(PDF)

**S8 Fig. Relative *Nacα* mRNA expression measured by Real-Time qPCR.** *Nacα* mRNA levels are reduced in adult hearts of HTT flies subject to *Nacα* KD during pupal stages only. *Nacα* mRNA levels are also reduced in early pupal hearts of HTT flies subject to *Nacα* KD during embryonic stages only.
(PDF)

**S9 Fig. *Nacα* and *Hox* genes interact to redirect differentiation of Multipotent Cardiac Progenitors (MCPs). A**, Knockdown (KD) of *Nacα*, *Hox* genes or their combination did not produce a significant change in the proportion of endothelial cell (CDH5+). The KD of transcription factors *Gata4/6*,*MyoCD* increased the proportion of endothelial cells. **B**, Representative images of immunohistological staining for select conditions.
(PDF)

**S10 Fig. Images of fluorescently labeled (tdtK) larval aorta and heart (abdominal segments 3 and 4) of White Pupae.** Heart tubes were present prior to cardiac remodeling with any of the *SRP* subunit knocked down using Hand4.2-GAL4, suggesting *SRP* subunits influence cardiac remodeling into adult structures during metamorphosis. Internal valves separating the

larval aorta and heart are marked by arrowheads.
(PDF)

**S1 Table. List of transgenic lines used in the study.** * indicates RNAi lines displayed in Figures. BDSC- Bloomington Drosophila Stock Center. VDRC- Vienna. Drosophila Resource Center.
(PDF)

**S1 Video. In vivo imaging of cardiac remodeling during pupation in control flies.** RFP driven by a heart-specific enhancer (tdtk) was imaged throughout pupation every 2 mins starting at approximately 6–8 hour APF (After Puparium Formation). The larval aorta and heart were visualized by RFP signal during early pupation. As the heart remodeled, RFP signal declined, especially in the larval aorta (anterior segments of the heart tube). As remodeling continued, RFP signal returned in the heart, and adult heart structures were evident, including wider heart diameters, formed ostia, and valves. Timestamps indicate APFs. Development of the heart may be slowed due to reduced ambient temperatures in the microscope room. See Fig 1 for still images from the video.
(MP4)

**S2 Video. In vivo imaging of cardiac remodeling during pupation in Nacα knockdown flies.** RFP driven by a heart-specific enhancer (tdtk) was imaged throughout pupation every 2 mins starting at approximately 6–8 hour APF (After Puparium Formation). The larval aorta and heart were visualized by RFP signal during early pupation. As the heart remodeled, RFP signal declined, especially in the larval aorta (anterior segments of the heart tube). As remodeling continued, RFP signal did not return, and no adult heart structures were evident, indicating complete lysis of the heart. Instead, the midline was filled in with rounded fat cells, similar to bic knockdown. Timestamps indicate APFs. Development of the heart may be slowed due to reduced ambient temperatures in the microscope room. See Fig 1 for still images from the video.
(MP4)

**S3 Video. In vivo imaging of cardiac remodeling during pupation in bic knockdown flies.** RFP driven by a heart-specific enhancer (tdtk) was imaged throughout pupation every 2 mins starting at approximately 6–8 hour APF (After Puparium Formation). The larval aorta and heart were visualized by RFP signal during early pupation. As the heart remodeled, RFP signal declined, especially in the larval aorta (anterior segments of the heart tube). As the heart remodeled, RFP signal declined, especially in the larval aorta (anterior segments of the heart tube). As remodeling continued, RFP signal did not return, and no adult heart structures were evident, indicating complete lysis of the heart. Instead, the midline was filled in with rounded fat cells, similar to Nacα knockdown. Timestamps indicate APFs. Development of the heart may be slowed due to reduced ambient temperatures in the microscope room. See Fig 1 for still images from the video.
(MP4)

## Author Contributions

**Conceptualization:** Analyne M. Schroeder, Rolf Bodmer.

**Data curation:** Analyne M. Schroeder.

**Formal analysis:** Analyne M. Schroeder.

**Funding acquisition:** Rolf Bodmer.

**Investigation:** Analyne M. Schroeder, Tanja Nielsen, Michaela Lynott.

**Methodology:** Analyne M. Schroeder, Georg Vogler, Alexandre R. Colas, Rolf Bodmer.

**Supervision:** Alexandre R. Colas, Rolf Bodmer.

**Validation:** Analyne M. Schroeder.

**Visualization:** Analyne M. Schroeder.

**Writing – original draft:** Analyne M. Schroeder, Rolf Bodmer.

**Writing – review & editing:** Analyne M. Schroeder, Georg Vogler, Alexandre R. Colas, Rolf Bodmer.

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
