## [Decision Letter · Decision Letter 0]

12 Apr 2022

Dear Dr Schroeder ,

Thank you very much for submitting your Research Article entitled 'Nascent Polypeptide Associated Complex and Signal Recognition Particle have cardiac-specific roles in heart development and remodeling' to PLOS Genetics.

The manuscript was fully evaluated at the editorial level and by independent peer reviewers. The reviewers appreciated the attention to an important problem, but raised some substantial concerns about the current manuscript. Based on the reviews, we will not be able to accept this version of the manuscript, but we would be willing to review a much-revised version. We cannot, of course, promise publication at that time.

If you decide to revise the manuscript for further consideration at PLOS Genetics, please aim to resubmit within the next 60 days, unless it will take extra time to address the concerns of the reviewers, in which case we would appreciate an expected resubmission date by email to plosgenetics@plos.org.

[LINK]

We are sorry that we cannot be more positive about your manuscript at this stage. Please do not hesitate to contact us if you have any concerns or questions.

Yours sincerely,

Pablo Wappner

Associate Editor

PLOS Genetics

Scott Williams

Section Editor: Human Variation

PLOS Genetics

Reviewer's Responses to Questions

**Comments to the Authors:**

Reviewer #1: The manuscript from Schroeder et al describes a novel role for the protein biogenesis cofactors Nascent polypeptide-Associated Complex (NAC) in fly and human iPSCs cardiac differentiation and morphogenesis. The authors show that knockdown (KD) of the alpha-subunit of NAC in the developing fly heart disrupts cardiac morphogenesis. KD of Naca or Nacb results in a fly with no heart. Indeed, Naca KD using specific cardiac GAL4 driver led to histolysis of the heart during metamorphosis. Temporal reduction of Naca expression during different developmental stages suggests that Naca is primarily requires developmentally for establishing a normal heart in adult, rather than maintaining its function in ages. They then identify that reduction of NACA in human iPSC-derived multipotent cardiac progenitors reduces the proportion of cardiomyocytes, whereas proportion of fibroblast is increased. To better understand the mechanisms driving complete histolysis of the heart during larval/pupal metamorphosis, the authors examine the expression of Hox gene Abd-B as this gene is normally expressed in the posterior larval heart region destined for histolysis. They show that Abd-B is misexpressed in KD Naca and that heart loss can be rescued by combined KD of Naca and Abd-B. Consistently, combined KD of NACA and HOXC12 or HOXD12 (Abd-B homologs) reverses the proportion of cardiomyocyte and fibroblast populations observed in KD NACA human iPSC-derived cells experiments.

In general, the study is well executed. The major scientific concern is that while the authors propose that reduction of Naca/NACA leads to misexpression of Abd-A/HOXC12 in cardiac cells, the molecular mechanism leading to this abnormal regulation remains unclear. The use of a cardiac specific Gal4 driver in fly to KD Naca only in the forming heart provides some support for the link between Naca and Abd-B during development, but there is no data confirming the specificity of this link in other cell types. It is easy to address with another drivers in order to understand if it is a generic or tissue specific function of Naca.

Data with tinC�4-Gal4 suggest that expression thresholds of Naca is required in cardiac cells. However, the authors did not address this point by quantifying Naca expression.

KD of Naca in the heart led to ectopic expression of Abd-A within the cell. The authors should discuss this point in term of sequence conservation. In the same perspective the authors should discuss the specificity of Naca on Abd-A and not other Hox genes.

Does temporal requirement of Naca is only associated with Abd-A misexpression?

The authors used cardiomyocytes derived from human iPSCs to determine whether similar role of NACA could be observed in other model system. It is clear that NACA siRNA inhibits CM differentiation, but the authors should convince the reader that progenitor cells are not blocked in their initial state.

siRNA against HOXC12 and HOXD12 can reverse the effect of siRNA NACA in the iPSC model. What is the specificity of this effect? Quantification of HOXC12 and HOXD12 in treated iPSCs should be evaluated too.

Temporal differentiation of iPSC to CM/fibroblasts should be also determine since premature differentiation of cardiac progenitors has been reported in Hoxb1 mutant mice.

Reviewer #2: The manuscript by Schroeder et al presents an investigation of the role of the nascent polypeptide associated complex (Nac) and, secondarily, the signal recognition particle (SRP) in cardiac development. Evidence from human genetics and previous work in Drosophila suggests a role for NACA, which encodes one of the two Nac subunits, in congenital heart disease. To investigate this further, Schroeder et al expressed shRNAs targeting each Nac subunit, Nac-alpha and bicaudal, and found that these were required for heart development. They extended this finding in two interesting ways. First they investigated the effects of Nac knockdown on Hox gene expression. In normal development, the larval heart region that is destined for histolysis during pupal development expresses Abd-B, while the larval aorta which remodels to become the adult heart expresses Ubx. They demonstrate that Nac-alpha knockdown produces ectopic expression of Abd-B, and histolysis of the entire heart in early pupal stages, and furthermore that heart development could be restored to the Nac-alpha knockdown by also knocking down Abd-B. Secondly, using the GAL4-GAL80ts system they demonstrate a specific temporal requirement for Nac-alpha activity during embryonic and pupal stages.

I found this part of the manuscript to be convincing, with only minor concerns that I will outline below.

The authors next investigate, by treating human iPSCs with siRNAs, whether the same regulatory interaction between Nac-alpha and Hox genes is involved in cardiomyocyte differentiation. There are two apparent Abd-B orthologues in humans, HOXC12 and HOXD12. In these cells, knockdown of Nac-alpha reduces the proportion of cardiomyocytes and increases that of fibroblasts. Knockdown of either HOXC12 or HOXD12 on their own had little effect. In combination with Nac-alpha knockdown, knockdown of either HOXC12, HOXD12, or both, partially rescued the effects of Nac-alpha knockdown to roughly equivalent extents.

For me, this result is weak because the double knockdown of HOXC12 and HOXD12 is not examined on its own. It is possible that these genes have redundant functions and the double knockdown would affect the proportions of the various cell types. Also, to further examine whether the regulatory pathway in flies is conserved in humans, the effects of overexpression of HOXC12/HOXD12 should be examined.

The final figure examines the effects of disrupting SRP components on cardiac development. This is of interest because SRP, like Nac, is also recruited to nascent polypeptides. The authors demonstrate that knockdown of various SRP subunits produces a variety of phenotypes, but these results are not extended to the level of any mechanistic insights.

Further comments and concerns:

Fig 1D-F: I find it very hard to see the described larval heart phenotypes in these panels. The data are more convincing in Fig S1A. The authors should consider improving the imaging in Fig 1 or substituting the data in Fig S1 to demonstrate the existence of larval hearts in all genotypes.

Fig S2: The nature of the control should be provided.

Discussion: This is very long and poorly organized, moving from flies to humans to mice, then back to flies, etc. I think it could be reduced by 50% without substantial loss. The paragraph starting with line 476 includes a fair bit of reiteration of results, the paragraph starting with line 492 could be deleted, and the two paragraphs that discuss Figure 6 are speculative and go far beyond the supporting experimental data. Finally, the presentation of the role of the Nac complex in embryonic patterning in the introduction and discussion contains inaccuracies. Nac is implicated in RNA localization which in turn is required for oskar and nanos translation, so it is not necessarily the case that its effects are only at the level of translation. Relevant to lines 465-468, Oskar protein begins to accumulate at the posterior pole already during mid-oogenesis, not just in the syncytial embryo as this discussion implies.

Reviewer #3: In this manuscript the authors investigate the role of Nascent polypeptide-Associated Complex (NAC) and Signal-Recognition-Particle (SRP) during heart development of Drosophila. Interestingly the authors suggest a patterning role for NAC and SRP rather than a general translational regulation role for these two important complexes.

Their data show that two NAC subunits (Nac-alpha and bicaudal/Nac-beta) are needed for normal heart remodelling during pupal stages. The heart is present at early pupal stages but lost during remodelling in the pupa upon Nac-alpha or bic knock-down. Interestingly this phenotype is rescued by double knock-down of Nac-alpha and the Hox gene AbdB. The latter becomes ectopically expressed upon NAC knock-down. The authors show that absence of AbdB is normally needed for effective heart remodelling from the larval to the adult heart during pupal stages.

From temperature shift experiments the authors hypothesise that Nac-alpha is needed to ‘pre-program’ cardiomyocytes already in the embryo for their later remodelling during the pupal stage. How this pre-programming should occur remains mysterious.

Additionally, the authors investigate a role of SRP subunits and find phenotypes that suggest some spatially restricted function of SRP during heart development in certain segments.

While I find the manuscript overall interesting, I have a number of suggestions that should be addressed by the authors during revisions to substantiate or tone-down some of the major conclusions.

Important points:

1. RNAi specificity and genotypes. Importantly, the authors use two independent RNAi lines to knock-down Nac-alpha and bic, at least according to the methods. However, Figures 1 and 2 show only the phenotype of one of each without specifying which one was used. Please add the exact genotypes in the figure legend and provide the second RNAi line phenotype in the same figure or in the supplemental figure (at least for Figure 1).

2. The fact that mis-expression of AbdB in developing pupal hearts is induced as a consequence of NAC knock-down is really interesting. However, it is hard to see on the current image in Figure 3E. Please provide separate grey channel images for the AbdB and the DAPI stains in control and RNAi that we can appreciate the signal in the nuclei.

3. The authors put forward the interesting idea that NAC is needed specifically in the embryo to prime the heart for later remodelling. This argument largely rests on the finding that knock-down of Nac-alpha during pupal stages only does not result in a strong phenotype. However, it is a well-known phenomenon that knock-down by RNAi depends on the driver used and takes time to be complete. Hence, only knocking-down at pupal stages might simple not reduce the proteins levels of Nac-alpha enough to produce a phenotype. This is consistent with the finding of the authors that the strongest phenotype is found when Nac-alpha is continuously knocked-down or knocked-down in the embryo and in pupae. Thus, the authors may want to explore additional routes to back up their conclusion on this important point. Otherwise, the strength of the phenotype may simply correlate with proteins levels during the remodelling stage.

Possibilities would be to use an antibody against Nac-alpha to show its absence in remodelling pupal hearts following knock-down in pupa only, in contrast to its potential presence in pupae when knocked-down the first 24h or 48h of development only.

Alternatives could be using a protein degradation system such us deGradeFP.

4. Figure 5 is missing a control for effectiveness of NACA RNAi when combined with a second siRNA against an unrelated gene. Any of the combinations shown at the moment appear to rescue, even though this rescue is not very obvious in the overview images. Would higher magnifications help?

5. Can the different phenotypes seen after knock-down of the various SRP subunits not simply be explained by varying effectiveness of knock-down? Again, no second RNAi lines are shown and no protein amounts were measured, hence the shown phenotypes are very hard to interpret. In my opinion, it would take a significant amount of more work to support the hypothesis of the authors that the different SRP subunits do different things in the different heart cells. Properly investigating this hypothesis will take an entire manuscript on its own. How is the phenotype when the same RNAi lines were used in skeletal muscle cells?

6. The discussion is overly long.

Minor points.

1. The heart remodelling movies are really nice. Please include a time stamp in the movies. Compression of the 70MB large movie files would be useful.

2. Figure 2C and F, please shift the driver labels a bit to the left or change the organisation of the labels. Hard to grasp which RNAi was combined with which driver in which of the graphs. Please label micro with µ. What is a KK control? An unrelated RNAi line? Were no GD lines used?

**Have all data underlying the figures and results presented in the manuscript been provided?**

Reviewer #1: Yes

Reviewer #2: Yes

Reviewer #3: Yes

PLOS authors have the option to publish the peer review history of their article (what does this mean?). If published, this will include your full peer review and any attached files.

Reviewer #1: **Yes: **Stéphane Zaffran

Reviewer #2: **Yes: **Paul Lasko

Reviewer #3: No

---

## [Decision Letter · Decision Letter 1]

27 Sep 2022

Dear Dr  Schroeder,

We are pleased to inform you that your manuscript entitled "Nascent Polypeptide Associated Complex and Signal Recognition Particle have cardiac-specific roles in heart development and remodeling" has been editorially accepted for publication in PLOS Genetics. Congratulations!

Yours sincerely,

Pablo Wappner

Academic Editor

PLOS Genetics

Scott Williams

Section Editor

PLOS Genetics

Comments from the reviewers (if applicable):

Reviewer's Responses to Questions

**Comments to the Authors:**

Reviewer #1: The manuscript from Schroeder et al. has been well revised. The authors have completed significant revisions based on my comments. I find the current version of the manuscript considerably improved. The authors have replied to my main concern regarding the tissue specificity of Naca and Abd-B in the heart. The authors used several Gal4 lines to KD Naca in different cell types including pericardial cells. These novel data provide additional support for a tissue-specific effect of Naca KD in the heart. As requested, the authors have measured Naca mRNA levels at different stages of development, however they did decide to include the Naca expression data only in pupae and in newly eclosed adult hearts in the revision. In addition to Abd-B, the authors have included data examining the Hox gene abd-A. Based on their results the authors conclude that there is specificity in interaction between Naca and Hox genes. Cardiomyocytes derived from human iPSCs was used as human model. As requested, the authors used differentiation markers to show that cells are not stuck in a progenitor state. Quantification of HOXC12 and HOXD12 alone are included in the revision. The current version included a shorten discussion. Therefore, the authors have satisfactorily answered all my questions.

Reviewer #3: The authors have addressed all my initial suggestions in a very satisfactory way. I recommend publication of this interesting and well presented manuscript.

**Have all data underlying the figures and results presented in the manuscript been provided?**

Reviewer #1: Yes

Reviewer #3: Yes

PLOS authors have the option to publish the peer review history of their article (what does this mean?). If published, this will include your full peer review and any attached files.

Reviewer #1: No

Reviewer #3: No

**Data Deposition**

http://datadryad.org/submit?journalID=pgenetics&manu=PGENETICS-D-22-00273R1

**Press Queries**

---

## [Editor Report · Acceptance letter]

11 Oct 2022

PGENETICS-D-22-00273R1 

Nascent polypeptide-Associated Complex and Signal Recognition Particle have cardiac-specific roles in heart development and remodeling 

Dear Dr Schroeder, 

We are pleased to inform you that your manuscript entitled "Nascent polypeptide-Associated Complex and Signal Recognition Particle have cardiac-specific roles in heart development and remodeling" has been formally accepted for publication in PLOS Genetics! Your manuscript is now with our production department and you will be notified of the publication date in due course.

With kind regards,

Zsofi Zombor

PLOS Genetics

On behalf of:
